# Unveiling the Genomic Features and Biocontrol Potential of *Trichoderma hamatum* Against Root Rot Pathogens

**DOI:** 10.3390/jof11020126

**Published:** 2025-02-08

**Authors:** Yuzhou Feng, Xinyi Shuai, Jili Chen, Qing Zhang, Lijie Jia, Luzhi Sun, Yunxia Su, Yanyan Su, Gangqiang Dong, Tao Liu, Guangqiang Long

**Affiliations:** 1Key Laboratory of Medicinal Plant Biology, Yunnan Agricultural University, Kunming 650201, Chinashuaixinyi123@163.com (X.S.);; 2National and Local Joint Engineering Research Center on Germplasm Innovation & Utilization of Chinese Medicinal Materials in Southwest China, Yunnan Agricultural University, Kunming 650201, China; 3Amway China Botanical R&D Center, Wuxi 214115, China; winni.su@amway.com (Y.S.); tony.dong@amway.com (G.D.)

**Keywords:** *T. hamatum*, *Fusarium*, biological control, genome, root rot disease

## Abstract

*Fusarium* species are among the most significant pathogens causing root rot in *Panax notoginseng*. In this study, a strain of *Trichoderma hamatum* was isolated from the rhizosphere soil of *P. notoginseng* and subjected to whole-genome sequencing. Plate confrontation experiments were conducted to investigate the antagonistic effects of *T. hamatum* against *Fusarium oxysporum*, *Fusarium solani*, and *Fusarium acutatum*, the primary *Fusarium* species causing root rot. Whole-genome sequencing revealed 10,774 predicted genes in *T. hamatum*, of which 454 were associated with carbohydrate-active enzymes (CAZymes) involved in fungal cell wall degradation. Additionally, 11 biosynthetic gene clusters (BGCs) associated with antimicrobial production were identified, highlighting the biocontrol potential of *T. hamatum*. In plate confrontation experiments, *T. hamatum* showed substantial inhibition rates of 68.07%, 70.63%, and 66.12% against *F. oxysporum*, *F. solani*, and *F. acutatum*, respectively. Scanning electron microscopy suggested the hyperparasitism of *T. hamatum* against *F. solani*, which was characterized by spore production that adhered to the pathogen, thereby inhibiting its growth. These findings provide a theoretical foundation to enhance understanding of the biological control mechanisms of *T. hamatum*, supporting its potential applications in sustainable agriculture.

## 1. Introduction

*P. notoginseng* is a medicinal plant primarily cultivated in Wenshan Prefecture, Yunnan Province, China [1]. However, plant pathogens significantly limit the cultivation of *P. notoginseng*. Among these, root rot is the most destructive disease, leading to reduced yields or even crop failure, resulting in substantial losses for the *P. notoginseng* industry [2,3]. *Fusarium* and *Ilyonectria* are the primary pathogens responsible for root rot in *P. notoginseng* [4,5,6,7]. Chemical fungicides and soil fumigation are traditional approaches to controlling root rot. However, chemical residues can cause environmental pollution, pose risks to human health, and contribute to pathogen resistance [8,9]. Biocontrol agents are now recognized as a sustainable approach for the biological control of pathogens and the prevention of soil-borne diseases [10,11]. Probiotics can enhance bacterial community diversity and optimize network structure, thereby resisting pathogen invasion and promoting plant health [10,12]. However, the molecular and mechanistic basis of fungal suppression in biological control strategies remains poorly understood. Therefore, identifying the biocontrol fungi for soil-borne pathogens is crucial to establishing sustainable and effective biological control strategies. Among the promising biocontrol agents, species of the genus *Trichoderma* have gained attention for their ability to suppress soil-borne pathogens and promote plant health.

The genus *Trichoderma* belongs to the order *Hypocreales* of the phylum *Ascomycota*. *Trichoderma*, with its ability to utilize a wide range of nutrient sources, can successfully establish itself as a parasitic or multifunctional symbiotic organism. They thrive in the rhizosphere, soil, or any environment containing decaying plant material [13,14]. *Trichoderma* can inhibit plant pathogens through antagonistic activity or by stimulating the host plant’s defense responses [15,16]. The antagonistic mechanisms of *Trichoderma* include parasitism, the production of secondary metabolites, lytic enzyme activity, and ecological competition [15,17,18]. Parasitism is a typical behavior of Trichoderma, which produces hydrolytic enzymes to degrade the cell walls of host fungi, thereby parasitizing ascomycete fungi or closely related fungal species [19]. The production of secondary metabolites in *Trichoderma* includes antibiotic production and other non-ribosomal peptides, such as peptaibiotics, diketopiperazines, polyketides, and so on [18]. Lytic enzymes, including chitinases, glucanases, and proteases, are crucial for parasitism as they degrade host cell walls to acquire nutrients [20]. Moreover, *Trichoderma* suppresses pathogenic fungi by competing for space, nutrients, and attachment sites [21]. *Trichoderma* species, such as *T. harzianum ZC51* and *Trichoderma AAUW1*, effectively control plant diseases caused by *Fusarium*, with the latter achieving an 86.67% inhibition rate against *F. solani* [22,23]. However, *Trichoderma* exhibits different antagonistic abilities towards different pathogen species. For instance, *T. koningiopsis* 7819 exhibits strong antagonistic effects against *Rhizoctonia solani*, moderate effects against *F. graminearum* and *Neocosmospora rubicola*, and weaker effects against *F. oxysporum* [24]. Therefore, the antagonistic mechanisms of *Trichoderma* against various *Fusarium* species remain partially understood.

To date, more than 400 *Trichoderma* species have been identified and classified through molecular biology techniques [25]. Despite belonging to the same genus, the substantial genetic, phenotypic, and ecological diversity among Trichoderma species is reflected in their genomic variability [26]. Whole-genome sequencing of species such as *T. harzianum*, *T. asperellum*, and *T. atroviride* has enhanced our understanding of Trichoderma’s ecological and genetic characteristics, enabling comprehensive molecular analyses of biocontrol mechanisms [27,28,29,30]. Understanding the genomic structure of *Trichoderma* aids in elucidating the genetic and molecular mechanisms of fungal antagonism against *Fusarium* species that cause root rot in *P. notoginseng*. This study provides a theoretical foundation for advancing the biological control strategies of root rot in *P. notoginseng*.

This study aimed to investigate the biocontrol potential of *T. hamatum* against *Fusarium* species causing root rot in *P. notoginseng*. Specifically, *F. solani*, *F. oxysporum*, and *F. acutatum* were targeted as the primary pathogens. *T. hamatum* was isolated from the rhizosphere soil of healthy *P. notoginseng* plants and subjected to whole-genome sequencing and comparative genomic analysis to identify candidate homologous and unique genes. Plate confrontation assays were conducted to evaluate the differential antagonistic effects of *T. hamatum* on the targeted *Fusarium* species. Additionally, scanning electron microscopy was used to examine the physical interactions between *T. hamatum* and Fusarium species. Genomic analyses focused on genes encoding lytic enzymes and secondary metabolites to explore the molecular mechanisms underlying these interactions. The findings of this study aim to support the optimization of *T. hamatum* biocontrol efficacy in managing root rot and contribute to comparative genomic studies of *Trichoderma*, providing insights into biocontrol mechanisms and the evolutionary dynamics of *Trichoderma* isolates.

## 2. Material and Methods

### 2.1. Soil Collection

Soil samples were collected from the Wenshan Miaoxiang *P. notoginseng* Technology Experimental Field in October 2022. The plot chosen for sampling was land where *P. notoginseng* had been continuously cultivated, and no fungicides were used during the growth period, allowing the plants to grow naturally. Healthy plants were selected from a field predominantly affected by root rot. Three biological replicates were collected, each consisting of 4 to 6 healthy plants. The plants were carefully excavated to preserve root system integrity, and loose soil was gently shaken off. Adhering rhizosphere soil was then gently brushed off. Samples were immediately placed in iceboxes for preservation and stored at −80 °C upon arrival at the laboratory [31].

### 2.2. Isolation and Identification of Fungi

We employed the dilution spread plate method for fungal isolation. To prepare a serial dilution, 10 g of rhizosphere soil was mixed with 90 milliliters of sterile water. The resulting soil suspension was diluted to 10⁻⁵, and 100 μL of the diluted solution was inoculated onto potato dextrose agar (PDA) plates. Distinct fungal colonies were subsequently picked and purified by sub-culturing onto fresh PDA plates under sterile conditions. Pure cultures were obtained through hyphal tip transfer and repeated inoculations until contamination-free fungal growth was confirmed. Following isolation and purification, plates were incubated at 28 °C for 3–5 days. Isolated strains were preserved in 50% glycerol and stored at −80 °C. Genomic DNA was extracted from isolated fungal strains using a modified cetyltrimethylammonium bromide (CTAB) method. The internal transcribed spacer (ITS) region was amplified using primer pair ITS1/ITS4 [32]. PCR conditions included an initial denaturation at 94 °C for 10 min, followed by 35 cycles of denaturation at 94 °C for 30 s, annealing at 54 °C for 30 s, and extension at 72 °C for 30 s, with a final extension at 72 °C for 10 min. Sanger sequencing of PCR products was conducted by Sangon Biotech Co., Ltd. (Shanghai, China) using the same primers (ITS1/ITS4) as in PCR amplification. Sequences obtained were compared to ITS sequences in the NCBI database using BLAST for fungal species identification. In our preliminary studies, a total of 120 fungal strains were isolated, among which 20 were classified within the genus *Trichoderma*. For this study, we selected a specific strain of *T. hamatum* based on its pronounced antagonistic activity observed during initial screening.

### 2.3. Antagonistic Activity Against Plant Pathogenic Fungi

The control was single culture, and the antagonistic experiment was dual culture. The dual culture method was used to evaluate the antagonistic activity of a single *T. hamatum* strain against the plant pathogens *F. oxysporum*, *F. solani*, and *F. acutatum* on potato dextrose agar (PDA). Three independent biological replicates were conducted for each pathogen–antagonist combination, with each replicate performed in triplicate. The antagonistic assay measured the pathogen radius and the inhibition zone between the *T. hamatum* strain and the pathogen. Pathogenic and antagonistic fungi *T. hamatum* were inoculated on opposite sides of PDA plates. Plates with only pathogenic fungi served as controls (CK). Plates were incubated at 28 °C until the mycelium on control plates was fully grown, after which pathogen inhibition rates were measured. The percentage inhibition efficiency was calculated using (radius of the pathogen in CK—radius of the pathogen in tested strain)/radius of the pathogen in CK × 100%.

### 2.4. Scanning Electron Microscopy

The objective of using SEM was to examine the interactions between *T. hamatum* and the pathogens, focusing on structural changes in pathogen hyphae and evidence of hyperparasitism or inhibition. The three pathogenic fungi were cultured on potato dextrose agar (PDA) plates, serving as the control group (CK). For the experimental group, *T. hamatum* and the pathogenic fungi were co-cultured on PDA plates. A 0.5 cm diameter colony sample, including agar, was excised from the interaction zone where *T. hamatum* and pathogen mycelia overlapped. Fixed samples were washed three times with 0.1 M phosphate-buffered saline (PBS, pH 7.4), with each wash lasting 15 min. Samples were then fixed in 1% osmium tetroxide in 0.1 M PBS (pH 7.4) at room temperature and protected from light for 1–2 h. The samples were washed three times with 0.1 M PBS (pH 7.4), each wash lasting 15 min. Samples were then dehydrated through a graded ethanol series (30%, 50%, 70%, 80%, 90%, 95%, and twice at 100%), each step lasting 15 min, followed by 15 min in isoamyl acetate. The samples were then dried in a critical point dryer. Samples were then adhered to conductive carbon double-sided tape and placed on a sample holder for gold coating for approximately 30 s in an ion sputter coater. Samples were examined and photographed using a scanning electron microscope (Hitachi SU8100, Tokyo, Japan).

### 2.5. DNA Extraction and Sequencing

The purpose of DNA extraction and sequencing was to analyze the genomic features of *T. hamatum* that contribute to its biocontrol potential against root rot pathogens. Collected mycelium was used to extract high-quality DNA with the QIAGEN^®^ Genomic DNA Kit. DNA concentration and quality were assessed using a NanoDrop 2000 spectrophotometer (NanoDrop Technologies, Wilmington, DE, USA), a Qubit 3.0 Fluorometer (Life Technologies, Carlsbad, CA, USA), and 0.8% agarose gel electrophoresis. Once DNA purity, concentration, and integrity were confirmed, libraries were constructed. Library fragment sizes were evaluated with an Agilent 2100 Bioanalyzer (Agilent Technologies, Santa Clara, California, USA). After library preparation, sequencing was performed on the PacBio Sequel II system using the SMART sequencing method. Raw data were filtered with fastp (version 0.21.0), and genome size and heterozygosity rate were estimated via a k-mer-based analysis. The longest high-quality genomic regions were identified using SMRT Link (version 7.0) software and the HQRF (High-Quality Region Finder) tool. Low-quality regions were filtered based on the Signal-to-Noise Ratio (SNR). Reads exceeding 1000 bp, a threshold ensuring sufficient read length for downstream analyses, were selected as quality-controlled sequencing data.

### 2.6. Genome Assembly

The initial genome assembly was performed using Hifiasm (version 0.13) on third-generation sequencing data. Subsequently, second-generation sequencing with Illumina NovaSeq was conducted, and the genome was polished with four rounds of NextPolish, resulting in the corrected (polished) genome. To evaluate assembly quality, genome completeness was assessed with BUSCO (version 5.2.2) based on the fungi_odb10 database. Second-generation sequencing data were aligned to the genome with BWA (version 0.7.17). Alignment statistics were calculated using SAMtools (version 1.4) and BCFtools (version 1.8.0) [33]. Homozygous sites were identified as genomic error sites to calculate the genome’s single base error rate.

### 2.7. Gene Prediction and Annotation

Microsatellite loci were identified in the sequences using GMATA (version 2.2). RepeatMasker (version open-1.0.11) was used to identify transposable element (TE) repeat sequences in *T. hamatum* using a curated repeat sequence database. Gene structure was predicted using PASA (version 2.3.3), GeMoMa (version 1.6.1), and AUGUSTUS (version 3.3.1). The predictions were integrated using Evidence Modeler (version 1.1.1). TransposonPSI was then used to remove genes containing transposable elements and incorrect coding, yielding the final gene set. Non-coding RNA (ncRNA) was predicted using Infernal (version 1.1.2) by aligning sequences to the Rfam database. tRNAs were identified with tRNAscan-SE (version 2.0), and rRNAs and their subunits were identified with RNAmmer (version 1.2). Gene functions were annotated by aligning genome protein sequences to the Non-Redundant Protein (NR), KEGG, Swiss-Prot, and KOG databases using Blastp (version 2.9.0).

### 2.8. Statistical Analysis

For the antagonistic activity assays. All experiments were conducted in triplicate to ensure reproducibility, and the results are presented as the mean ± standard deviation (SD). Statistical analyses were performed using SPSS v27.0.

## 3. Results

### 3.1. The Antagonistic Activity

The *T. hamatum* strain isolated from the rhizosphere soil of healthy *P. notoginseng* plants demonstrated significant inhibitory effects against *F. solani*, *F. oxysporum*, and *F. acutatum* in dual confrontation experiments. The inhibition rates of *T. hamatum* against these three *Fusarium* species were 68.07%, 70.63%, and 66.12%, respectively (Appendix A). In its confrontation with *F. solani*, *T. hamatum* completely overgrew the pathogen’s mycelium and produced numerous green spores, which may suggest a parasitic interaction, although the direct mechanism remains to be confirmed (Figure 1D). For *F. oxysporum* and *F. acutatum*, inhibition zones were observed, appearing transparent and diffuse, respectively (Figure 1E,F). These findings indicate that *T. hamatum* likely employs multiple mechanisms, including the potential secretion of antifungal compounds, to inhibit these pathogens.

### 3.2. Genomic Characteristics of T. hamatum

HiFi sequencing of the *T. hamatum* was performed, generating 6.82 Gb of raw data with an average sequencing depth of 162.6×. The sequencing produced 393,002 high-quality reads after filtering (Appendix A). The Hi-C-assisted genome assembly resulted in a total genome length of 41.93 Mb, an N50 value of 6.07 Mb, and 31 contigs. The GC content of the assembled genome was 46.37% (Figure 2, Appendix A). Genome alignment showed an alignment rate of 99.00% with a coverage of 99.98%, indicating a high-quality assembly. BUSCO analysis further demonstrated that 98.7% of core conserved genes were complete, highlighting the genome’s completeness and integrity for downstream analyses (Appendix A).

### 3.3. Functional Genes Annotation

Gene prediction analysis identified a total of 10,774 genes, with an average length of 1673.37 bp and an average protein-coding gene length of 1537.48 bp. On average, each gene contained 2.73 exons, with an average exon length of 563.32 bp and an average intron length of 117.05 bp (Appendix A). Annotation of non-coding RNAs (ncRNAs) revealed 400 distinct ncRNAs, including 221 tRNAs, 338 rRNAs, and 25 snRNAs (Table 1).

Gene annotation was conducted using InterPro, GO, KEGG, SwissProt, TrEMBL, and NR databases (Appendix A). Functional annotation through the GO database classified 7922 genes (73.53% of all genes) into three main categories: biological processes (6557 genes), cellular components (5130 genes), and molecular functions (9303 genes). Molecular functions constituted the largest category, with primary roles including catalytic activity, protein binding, and transporter activity (Appendix A). KEGG pathway analysis identified 4164 genes (38.65%) mapped to five major pathways, with metabolism-related genes being the most abundant, particularly those involved in carbon, amino acid, and lipid metabolism (Appendix A). Additionally, 37 genes were identified as being involved in the biosynthesis of secondary metabolites (Appendix A).

### 3.4. Homology Analysis and Phylogenetic Analysis

The phylogenetic relationship of the *T. hamatum* strain was evaluated based on Internal Transcribed Spacer (ITS) sequences, comparing it with ten other *Trichoderma* species and two outgroup fungi. The analysis utilized 2378 shared single-copy homologous sequences, providing a robust framework for assessing genetic similarities and evolutionary relationships (Figure 3). The ITS sequence clusters *T. atroviride*, *T. gamsii*, *T. hamatum XP3S-3oo*, *T. hamatumX2zz*, *T. asperellum*, and *T. hamatum* into a single branch. Similarly, single-copy homologous sequences group *T. gamsii*, *T. atroviride*, *T. asperellum*, *T. hamatum GD12*, *T. hamatum FBL587*, and *T. hamatum* into one branch. The phylogenetic tree that the *T. hamatum* strain forms a distinct clade closely related to *T. hamatum GD12* and *T. hamatum FBL587* strains, confirming its taxonomic placement within the *T. hamatum* species. Analysis of orthologous relationships among strains revealed that no paralogous genes were detected in the *T. hamatum* strain, and the number of genes not involved in clustering was minimal. Furthermore, its gene structure closely resembles that of the *T. hamatum GD12* strain reported in previous studies. (Appendix A). These findings suggest that the *T. hamatum* strain has undergone relatively stable evolution with highly specialized gene functions.

Phylogenetic trees were constructed using the eleven *Trichoderma* species and two outgroup strains. The average divergence time for the outgroup represented by *Aspergillus fumigatus* is estimated to be 187.8 million years ago (MYA). This predates the diversification of Hypocreaceae and Xylariaceae. Within the *Trichoderma* clade, the divergence from *Xylaria* occurred at approximately 141.7 MYA. Within Hypocreaceae, the split between *T. hamatum* and other species occurred around 7.1 MYA. *T. hamatum* diverged from *T. hamatum FBL587* and *T. hamatum GD12* relatively recently, around 0.9 million years ago, while the strains *T. hamatum FBL587* and *T. hamatum GD12* diverged relatively recently at 0.3 MYA. Specifically, the strain of *T. hamatum* examined in this study diverged from *T. hamatum GD12* at approximately 0.9 MYA (Appendix A). During the evolution of the involved thirteen fungal species, gene family contractions occurred more frequently than gene family expansions. The *Trichoderma* species, including *T. virens* and *T. harzianum*, demonstrated a higher level of gene family expansion 186 and 174, respectively, compared to other species in the clade. Notably, *T. hamatum* experienced significant evolutionary changes in 332 gene families, comprising 163 expansions and 196 contractions (Appendix A). Gene expansion and contraction can provide insights into an organism’s evolutionary history and functional adaptations within the organism.

### 3.5. Anzyme Production Capability of T. hamatum

Carbohydrate-active enzymes (CAZymes) play a significant role in various biological processes within the genus *Trichoderma*, particularly concerning fungal parasitism [27]. A total of 454 genes encoding CAZymes were identified based on the dbCANseq database. All genes were classified into six categories: 259 glycoside hydrolases (GHs), 13 carbohydrate-binding modules (CBMs), 70 auxiliary activities (AAs), 23 carbohydrate esterases (CEs), 78 glycosyltransferases (GTs), and 11 polysaccharide lyases (PLs) (Figure 4). Among these families, glycoside hydrolases (GHs) are the most abundant, with the 4 largest gene families being GH18 (28 genes), GH3 (19 genes), GH2 (10 genes), and GH27 (9 genes). These genes are associated with various enzymatic activities. Additionally, 8 genes from the GH55 family encode exo-β-1,3-glucanase, and 6 genes from the GH75 family encode chitosanase, further emphasizing the hydrolases synesis function of the fungi (Appendix A). These enzymes facilitate the breakdown of structural components in fungal cell walls, supporting the parasitic activity of *T. hamatum*. Collectively, the presence of these hydrolase genes highlights the significant potential of *T. hamatum* as a mycoparasitic biocontrol agent.

### 3.6. Biocontrol-Related Secondary Metabolite Biosynthetic Gene Clusters

The antiSMASH 7.1.0 tool was employed to predict secondary metabolite biosynthetic gene clusters (BGCs) within the *T. hamatum* genome. A total of 52 biosynthetic gene clusters (BCGs) distributed across eight contigs were predicted (Appendix A). Among these, 26 BCGs related to non-ribosomal peptide synthetases (NRPS and NRPS-like), 16 BCGs encoded genes related to type I polyketide synthases (T1PKS), and 7 BCGs were terpene synthases. To assess the biocontrol potential of *T. hamatum*, BGCs associated with eleven types of secondary metabolites were identified (Appendix A). The structural formulas of secondary metabolites and their genetic region are shown in Figure 5. Three genes exhibited 100% amino acid sequence homology with known secondary metabolic secretion gene clusters were further analyzed. Region 2.6 of Ptg2 encodes NRPS genes involved in the synthesis of peramine (Figure 5j). Region 6.5 of Ptg6 encodes terpene-related genes related to the synthesis of clavaric acid (Figure 5k). Region 5. 2 in Ptg5 were NRPS-like genes and involved in the synthesis of choline (Figure 5g). Additionally, two genes display 75% amino acid sequence similarity. The genes in region 1.6 (Pgt1, NRPS) related to trichoxide synthesis (Figure 5a), and region 4.3 (Pgt4 and T1PKS) related to the synthesis of dimerumic acid 11-mannoside and dimerumic acid (Figure 5e, f). Other BGCs are responsible for the biosynthesis of aurofusarin (Figure 5b), melinacidin IV (Figure 5c), leucinostatin A (Figure 5d), leucinostatin B (Figure 5h), and BII-rafflesfungin (Figure 5i).

### 3.7. Characterizing the Interactions of T. hamatum with Fusarium Pathogens

Scanning electron microscopy (SEM) revealed distinct morphological responses of pathogenic fungi in co-culture with *T. hamatum*. When *F. solani* was grown alone, its hyphae appeared smooth and intact, with no evidence of sporulation (Figure 6A). In co-culture conditions, *T. hamatum* conidia were observed adhering to *F. solani* hyphae in the interaction zone (Figure 6D). Similarly, in the absence of the antagonistic strain, the hyphae of *F. oxysporum* and *F. acutatum* were smooth and regular (Figure 6B,C). However, in the presence of *T. hamatum*, *F. oxysporum* hyphae exhibited pronounced dissolution and lysis (Figure 6E), while *F. acutatum* hyphae developed irregular wrinkles and interwoven mycelial masses (Figure 6F). These findings indicate that *T. hamatum* employs distinct inhibitory mechanisms against the three pathogenic fungi, as evidenced by the varied morphological alterations observed.

## 4. Discussion

Biological control, as an environmentally friendly approach to disease prevention and management, reduces reliance on chemical pesticides and mitigates the environmental pollution associated with their use. It has been widely applied in crop cultivation. However, root rot disease caused by *Fusarium* in the continuous cropping of *P. notoginseng*, a traditional Chinese medicinal plant, remains a significant challenge. This study investigated the antagonistic strain *T. hamatum*, isolated from the rhizosphere soil of *P. notoginseng*, for its potential to combat root rot disease. The strain demonstrated significant antagonistic activity against the primary pathogens responsible for *P. notoginseng* root rot, including *F. oxysporum*, *F. solani*, and *F. acutatum* (Appendix A). *Trichoderma* plays a significant role in global agricultural production and serves as an important resource for biological control [15,34].

This study analyzed the genomic structure of the biocontrol fungus *T. hamatum* through genome sequencing and conducted a comparative genomic analysis with ten other *Trichoderma* species, as well as two outgroup species. The phylogenetic tree revealed that *T. hamatum* is most closely related to *T. hamatum GD12* and *T. hamatum FBL587*, confirming its taxonomic placement within the *T. hamatum* species (Figure 3). The observed gene family expansions (163 genes) and contractions (196 genes) suggest significant genomic restructuring, potentially contributing to ecological adaptation and functional specialization. The absence of paralogous genes and minimal unclustered genes further indicates genomic stability. Paralogous genes are a subclass of homologous genes generated through gene duplication [35]. Their formation represents a critical mechanism for enhancing genetic diversity during genome evolution. The previously studied *T. hamatumGD12* strain has been well-documented for its effectiveness in promoting plant growth (PGP) and its strong biological control capabilities [36]. The *T. hamatum* strain analyzed in this study exhibits a high degree of similarity to *T. hamatum GD12* in gene structure (Appendix A), suggesting its significant potential for biological control applications. In addition, The number of genes in biocontrol *Trichoderma* species (>10,000) is greater than that in industrial *Trichoderma* species (<10,000) [28]. The *T. hamatum* strain studied in our research possesses 10,774 genes (Appendix A), suggesting a greater inclination toward biocontrol applications.

The direct biological control mechanisms employed by *Trichoderma* against plant pathogens include parasitism, antibiotic production, the action of lytic enzymes, competition for ecological niches, and resource competition [15,37]. These molecular mechanisms are associated with multiple genes in *Trichoderma* [38]. The plate confrontation assay and scanning electron microscopy demonstrated that *T. hamatum* employed distinct strategies to inhibit pathogenic fungi causing root rot in various *Fusarium* species. The antagonistic mode against *F. solani* involved parasitism, with *F. solani* being targeted through sporogenic parasitism. In contrast, *F. oxysporum* was inhibited via hyphal lysis, highlighting a different mode of antagonism. The inhibition zones observed on the plate suggest that *T. hamatum* likely produces antibiotics or specific metabolites that inhibit the growth of the pathogenic fungi *F. oxysporum* and *F. acutatum* (Figure 1 and Figure 6). Moreover, the abundant spores of *Trichoderma* enable it to disperse efficiently and establish dominance within ecological niches, thereby enhancing its competitive ability against other microorganisms [39]. The green spores are potentially linked to the polyketide synthase (PKS) encoding genes in *Trichoderma*. Pigment-forming PKSs play a dual role in conidiospore pigmentation and the biosynthesis of low molecular weight pigments, such as aurofusarin and bikaverin [40]. Low molecular weight pigment-forming PKSs are implicated in fungal defense, mechanical stability, and stress resistance [41]. Specifically, sixteen type I polyketide synthases were detected in *T. hamatum*, and aurofusarin was also identified in type I PKS (Region 1.8) (Appendix A, Appendix A). A homolog of the low molecular weight pigment-forming PKS, the *pks*4 gene in *T. reesei*, has been confirmed to be involved in the synthesis of aurofusarin and bikaverin. The pks4 gene influences the stability of the conidiospore cell wall and the antagonistic ability of Trichoderma against other fungi [42]. Therefore, PKS-encoding genes potentially play a crucial role in the antagonistic activity against pathogens. Genomic functional annotation, antagonistic experiments, and SEM analysis all suggest that *T. hamatum* has the potential to suppress pathogen growth through parasitism.

Fungal secondary metabolite biosynthetic gene clusters (BGCs) are responsible for synthesizing a wide array of secondary metabolites with specific biological functions [39]. In the genome of *T. hamatum*, a total of 11 BGCs were identified, which are associated with the production of various antibiotic and antibacterial compounds, including trichoxide, melinacidin IV, leucinostatin A/B, peramine, and clavariac acid. Trichoxide, produced by *Trichoderma*, exhibits strong antifungal activity [43,44]. Melinacidin IV is a secondary metabolite of certain fungi and is classified as an antibiotic [45]. Moreover, leucinostatin A and leucinostatin B have been reported as substances with a broad spectrum of biological activities, including antimalarial, antiviral, antibacterial, antifungal, antitumor activities, and phytotoxicity [46]. The significant inhibitory effect of *T. hamatum* on *F. oxysporum* and *F. acutatum* may be attributed to the secondary metabolites it produces. Furthermore, BGCs in *T. hamatum* are also involved in the synthesis of peramine and clavariac acid, compounds that are commonly associated with insecticidal activity [47,48]. Clavariac acid can also stimulate plant defense mechanisms or promote plant growth [49]. These secondary metabolites not only enable *T. hamatum* to inhibit the growth of pathogens but also potentially assist in the growth of host plants while enhancing their resistance to diseases and pests. The secondary metabolites produced by Trichoderma have the potential to develop new biopesticides and biofertilizers.

In the action of lytic enzymes in the CAZymes analysis of the *T. hamatum* genome, 28 genes were detected in glycoside hydrolase family 18 (GH18) and 8 genes in family 75 (GH75) (Appendix A). GH18 encompasses all fungal chitinases, which are enzymes that hydrolyze glycosidic bonds in chitin. Chitosanases from GH75 are involved in chitin degradation [20]. The fungal cell wall is a vital barrier that preserves cell integrity and supports survival, with chitin as a key structural component [20]. The biocontrol agent *Trichoderma* produces lytic enzymes that can degrade the cell walls of pathogens [20]. The main lytic enzymes produced by *Trichoderma* include chitinases, β-1,3-glucanases, and proteases. These enzymes not only degrade chitin and chitosan in the cell wall but also participate in the breakdown of the host fungus’ cell wall during parasitism [50,51]. Chitinases also play roles in plant development and defense against fungal pathogens [52,53]. Previous studies have shown that hyperparasitic fungi such as *T. atroviride*, *T. virens*, and the saprotrophic fungus *T. reesei* have 29, 36, and 20 genes in GH18, respectively, while other filamentous fungi typically possess 10 to 20 genes in GH18 [54]. Thus, *T. hamatum* demonstrates significant biocontrol potential as a parasitic fungus. Additionally, the analysis of *T. hamatum* also identified other enzymes, including β-glucosidase, β-galactosidase, α-galactosidase, and exo-β-1,3-glucanase (Appendix A). The chitinases, β-glucosidases, and mannosidases secreted by *Trichoderma* work together to control pathogens like *Rhizoctonia solani* and *Fusarium* spp. [55,56]. The hyperparasitic response of *T. harzianum ALL42* to pathogens is host-dependent, with variations in hyphal entanglement and secreted proteins observed [55]. Overall, the biocontrol mechanisms of *Trichoderma* can vary significantly depending on the specific host it interacts with.

## 5. Conclusions

This study isolated a *T. hamatum* strain with significant antagonistic activity against *F. oxysporum*, *F. solani*, and *F. acutatum*, the pathogens responsible for root rot in *P. notoginseng*. Whole-genome sequencing revealed a complete genome with 10,774 genes, including carbohydrate-active enzymes (e.g., GH18 chitinases) and secondary metabolite biosynthetic clusters (e.g., aurofusarin and melinacidin IV), highlighting its biocontrol potential. Antagonistic assays demonstrated inhibition rates of 70.63%, 68.07%, and 66.12% against *F. solani*, *F. oxysporum*, and *F. acutatum*, respectively. Scanning electron microscopy revealed distinct inhibitory mechanisms, including sporulating parasitism against *F. solani* and hyphal lysis against *F. oxysporum* and *F. acutatum*. These findings provide specific insights into the genetic and mechanistic basis of *T. hamatum*’s biocontrol activity, offering a robust foundation for developing sustainable biological control strategies in *P. notoginseng* cultivation.

## Figures and Tables

**Figure 1 jof-11-00126-f001:**
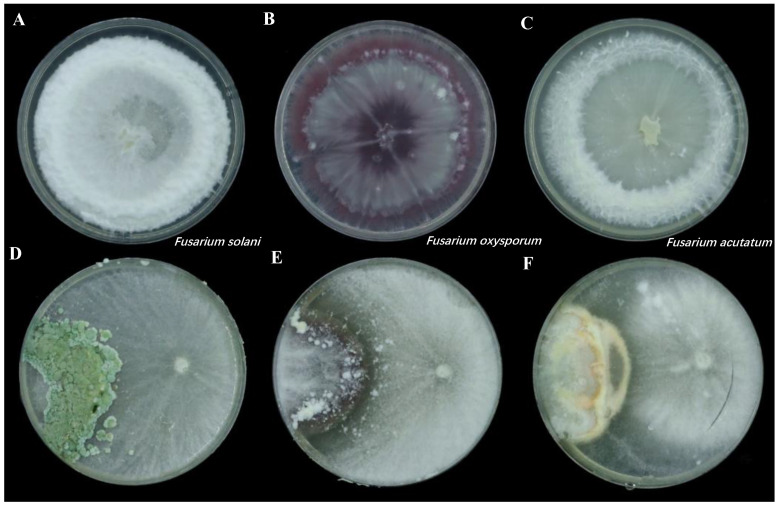
Images illustrating the inhibitory effects of *T. hamatum* on pathogenic fungi in antagonistic assays. Control cultures of *F. solani*, *F. oxysporum*, and *F. acutatum* cultured alone (**A**–**C**). Dual culture plates showing the pathogenic fungi inoculated on the left side, and *T. hamatum* inoculated on the right side (**D**–**F**).

**Figure 2 jof-11-00126-f002:**
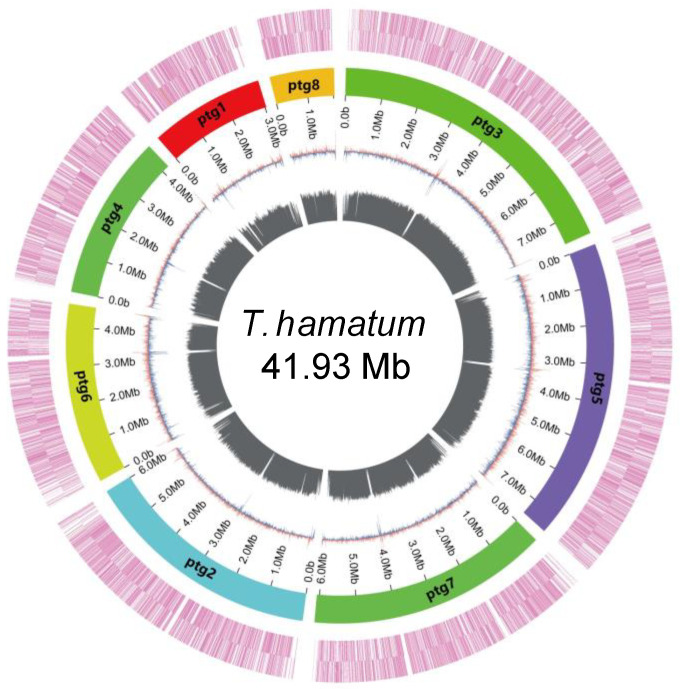
The Circos plot of the *T. hamatum* genome, with concentric circles representing GC skew, GC content, contig scaffolding, and gene density, respectively, from the innermost to the outermost circle.

**Figure 3 jof-11-00126-f003:**
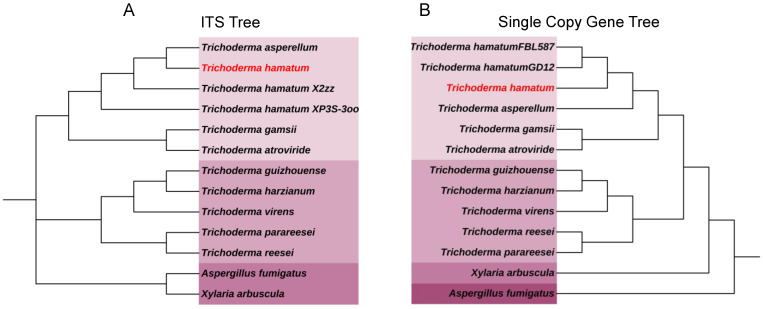
Phylogenetic dendrograms of *T. hamatum* based on (**A**) ITS sequences and (**B**) concatenated sequences of 2378 shared single-copy homologous genes.

**Figure 4 jof-11-00126-f004:**
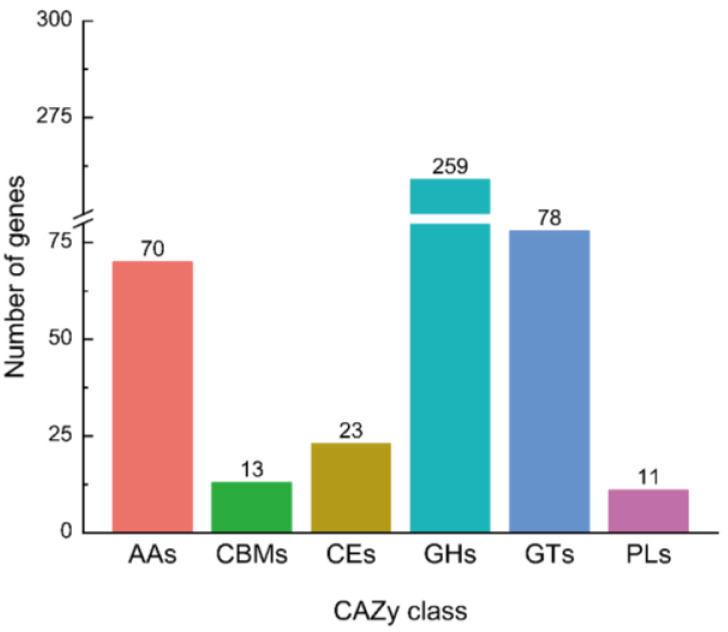
Classification of CAZy family genes in *Trichoderma hamatum*. AAs: auxiliary activities; CBMs: carbohydrate-binding modules; CEs: carbohydrate esterases; GHs: glycoside hydrolases; GTs: glycosyl transferases; PLs: polysaccharide lyases.

**Figure 5 jof-11-00126-f005:**
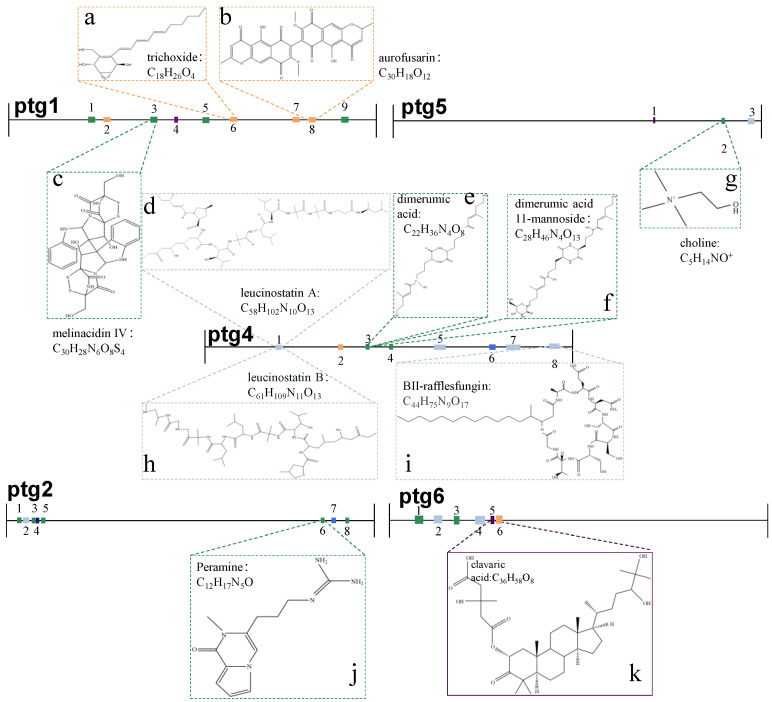
Secondary metabolite gene clusters identified in the *T. hamatum* Genome Using antiSMASH Software Version 7.0. (**a**) Trichoxide: C_19_H_18_O_4_, (**b**) Aurofusarin: C_30_H_28_O_12_, (**c**) Melinaidin IV: C_30_H_38_N_4_O_5_S_4_, (**d**) Leuconotatin A: C_54_H_102_N_10_O_13_, (**e**) Dimeric acid: C_22_H_36_N_4_O_6_, (**f**) Dimeric acid 11-mannoside: C_28_H_46_N_4_O_13_, (**g**) Choline: C_5_H_14_NO^+^, (**h**) Leuconostatin B: C_69_H_110_N_10_O_13_, (**i**) BII-rafflesfungin: C_44_H_75_N_5_O_17_, (**j**) Perramine: C_14_H_17_N_3_O, (**k**) Clavarinic acid: C_36_H_58_O_8_.

**Figure 6 jof-11-00126-f006:**
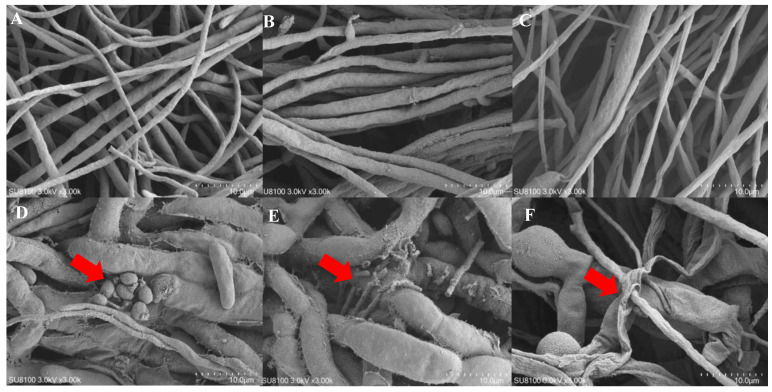
Scanning electron microscopy (SEM) images of *Fusarium* strains. Control cultures of *F. solani*, *F. oxysporum*, and *F. acutatum* without inoculation by *T. hamatum* (**A**–**C**). SEM images of cultures following inoculation with *T. hamatum* (**D**–**F**). The arrows indicate the interaction zone between *T. hamatum* and the pathogenic fungi, highlighting conidia adherence to pathogen hyphae and structural changes associated with fungal antagonism.

**Table 1 jof-11-00126-t001:** Statistics of non-coding RNAs.

Class	Type	Copy	Average Length (bp)	Total Length (bp)	% of Genome
miRNA	miRNA	0	0	0	0
tRNA	tRNA	221	87.90498	19,427	0.046331
rRNA	rRNA	338	2399.053	810,880	1.933838
	18S	146	1788.808	261,166	0.622845
	28S	141	3856.78	543,806	1.296903
	5S	51	115.8431	5908	0.01409
snRNA	snRNA	25	137.68	3442	0.008209
	CD-box	14	129.3571	1811	0.004319
	HACA-box	3	167	501	0.001195
	splicing	8	141.25	1130	0.002695
	scaRNA	0	0	0	0

## Data Availability

These data can be accessed through the NCBI BioProject PRJNA1154318 and BioSample SAMN43412261.

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
