# Peer review of "Unveiling the Genomic Features and Biocontrol Potential of Trichoderma hamatum Against Root Rot Pathogens"

_jof, 2025, doi:10.3390/jof11020126_

Round 1

Reviewer 1 Report

The manuscript (Jof-3344698) reports the genomic features and biocontrol potential of a Trichoderma hamatum strain isolated from the rhizosphere soil of P. notoginseng. In this study, the authors demonstrated the antagonistic effects of this strain of T. hamatum against Fusarium oxysporum, Fusarium solani, and Fusarium acutatum.
This is fine. However, I have raised some questions on the presentation of the manuscript that need to be addressed before the manuscript can be considered for publication.
From the review perspective, the authors should consider adequately describing the T. hamatum strain used in this study. No efforts are made in the manuscript to examine taxonomic relathionships of this strain in relation to other Trichoderma hamatum strains.

This is especially true with the availability of sequencing/genome data. Two T. hamatum genomes are publicy available at the genetic sequence database GenBank (NCBI). see

https://www.ncbi.nlm.nih.gov/datasets/genome/?taxon=49224. Based on genomic sequencing data of the T. hamatum strain used in this study, I invite authors to perform a comparative genomic analysis for selected Trichoderma hamatum strains.

Is the genome of the T. hamatum strain used in this study publicly available? It would be helpful to better clarify it. I invite authors to submit the nucleotide sequences of bacterial community used in this study in public genetic sequence database such as GenBank (NCBI). It will be easier for the readers to better follow the entire story.

I hope it helps.

see major comments

Author Response

Manuscript ID: jof-3344698

Title: Unveiling the genomic features and biocontrol potential of Trichoderma hamatum against root rot pathogens

Major comments:

The manuscript (Jof-3344698) reports the genomic features and biocontrol potential of a Trichoderma hamatum strain isolated from the rhizosphere soil of P. notoginseng. In this study, the authors demonstrated the antagonistic effects of this strain of T. hamatum against Fusarium oxysporum, Fusarium solani, and Fusarium acutatum. This is fine. However, I have raised some questions on the presentation of the manuscript that need to be addressed before the manuscript can be considered for publication.

Response:

Thank you for being able to provide valuable comments on the manuscript ID: jof-3344698 “Unveiling the genomic features and biocontrol potential of Trichoderma hamatum against root rot pathogens”, those comments are all valuable and very helpful for revising and improving our paper. We have carefully read the comments and revised the issues. In addition, we have resubmitted a new manuscript, and the modified part is highlighted in red, which is highlighted in blue in the response. If there are any questions in the manuscript, please don't hesitate to let us know. The following sections are our point-by-point responses.

  1. Comments: From the review perspective, the authors should consider adequately describing the T. hamatum strain used in this study. No efforts are made in the manuscript to examine taxonomic relathionships of this strain in relation to other Trichoderma hamatum strains.This is especially true with the availability of sequencing/genome data. Two T. hamatum genomes are publicy available at the genetic sequence database GenBank (NCBI). See https://www.ncbi.nlm.nih.gov/datasets/ genome/?taxon=49224. Based on genomic sequencing data of the T. hamatum strain used in this study, I invite authors to perform a comparative genomic analysis for selected Trichoderma hamatum strains.

Response:

We have successfully located the genome data for Trichoderma hamatum GD12 at NCBI. However, we were unable to find the genome annotation file for Trichoderma hamatum FBL 587 on NCBI. Despite extensive searches in other fungal genome databases, we could not obtain the genome annotation file for this strain. Therefore, we selected a closely related species, Trichoderma inhamatum, as an alternative for the comparative genomic analysis. Based on the results of this analysis, we have made the necessary revisions to the original text. Lines 264-294: “Analysis of orthologous relationships among strains revealed that no paralogous genes were detected in the T. hamatum strain, and the number of genes not involved in clustering was minimal. Furthermore, its gene structure closely resembles that of the T. hamatum GD12 strain reported in previous studies. (Figure S3). These findings suggest that the T. hamatum strain has undergone relatively stable evolution, with highly specialized gene functions. Phylogenetic analysis was conducted using the ITS sequences of the T. hamatum strain along with ten other Trichoderma species and two outgroup fungi. The analysis was based on 2,307 shared single-copy homologous sequences (Figure. 3). The ITS sequence clusters Trichoderma gamsii, T. atroviride, T. hamatum XP3S-3oo, T. asperellum, and T. hamatum into a single branch. Similarly, single-copy homologous sequences group T. gamsii, T. atroviride, T. asperellum, T. hamatum GD12, and T. hamatum into one branch. Phylogenetic analysis reveals that T. hamatum shares the closest evolutionary relationships with T. hamatum GD12 and T. asperellum.

Phylogenetic trees were constructed using the eleven Trichoderma species and two outgroup strains. The average divergence time for the outgroup represented by Aspergillus fumigatus is estimated to be 238.2 million years ago (MYA). Within the Trichoderma clade, the divergence from Xylaria occurred approximately 161.3 MYA. Among the Trichoderma species, T. gamsii, T. atroviride, T. hamatum, T. hamatum GD12, and T. asperellum diverged from T. parareesei, T. reesei, T. virens, T. guizhouense, T. inhamatum and T. harzianum at various times. Specifically, the strain of T. hamatum examined in this study diverged from T. hamatum GD12 approximately 1.7 MYA (Figure. S4).

During the evolution of the involved thirteen fungal species, gene family contrac-tions occurred more frequently than gene family expansions. Among the Trichoderma strains, T. gamsii, T. atroviride, T. asperellum, T. hamatum GD12, and T. hamatum exhibited 110, 375, 165, 65, and 119 gene family expansions, respectively, along with 416, 154, 193, 320, and 144 gene family contractions, respectively. Notably, T. hamatum experienced significant evolutionary changes in 263 gene families, comprising 119 expansions and 144 contractions (Figure. S5).”

  1. Comments: Is the genome of the T. hamatum strain used in this study publicly available? It would be helpful to better clarify it. I invite authors to submit the nucleotide sequences of bacterial community used in this study in public genetic sequence database such as GenBank (NCBI). It will be easier for the readers to better follow the entire story.

Response:

Thanks for your suggestions. We have uploaded our genome data to NCBI and incorporated a Data Availability Statement into the manuscript. The statement can be found on lines 475–476 of the revised version: “Data Availability Statement: These data can be accessed through the NCBI BioProject PRJNA1154318 and BioSample SAMN43412261.”

Reviewer 2 Report

Dear Editor and authors

 I have reviewed the manuscript ID: jof-3344698 entitled: “Unveiling the Genomic Features and Biocontrol Potential of Trichoderma hamatum Against Root Rot Pathogens.

First of all, I apologize if my English is not correct because my mother tongue is Spanish.

I consider that the work is very interesting because it addresses a very current subject as the search alternatives ecofriendly to reduce the use of agrochemicals to plant disease management. It is important to highlight that the research addresses the study of a Trichoderma species as biological control agents. In this sense, authors evaluated the antagonistic effect of T. hamatum isolated from the rhizosphere soil of Panax notoginseng against threes Fusarium species causing root rot.

On the other hand, the authors carried out genomics studies to elucidate the biocontrol mechanisms involved by the T. hamatum which is a contribution of knowledge of this topic.

However, I consider that the authors could improve the manuscript by improving and clarifying some aspects of materials and methods. Also, I think the work has some weaknesses in relation to antagonism tests and to the interpretation of the results and the mechanisms involved in the antagonism and they should improve it.

For example, it is not clear how many Trichoderma strains were initially isolated from  rhizosphere soil. It is also not clear how many strains were used for testing and identification.

With respect to the identification, it is widely known that the identification of Trichoderma species is very difficult and that more of ITS regions are required to determine at level species.

According to mentioned I recommend its publication in the Journal of Fungi after improvements to the manuscript have been made.

Dear Editor and authors

I am listed below other comments, suggestions that I think should be made in the text for publication in order to improve the manuscript.

Abstract

With respect this sentence: “Scanning electron microscopy revealed that T. hamatum inhibits pathogen growth through sporulating parasitism and secretion of secondary metabolites”.

How can the authors can assert the inhibition of the pathogen through secretion of secondary metabolites by scanning electron microscopy?

It is not clear the inhibition of the pathogen growth through sporulating parasitism……

See the comment below (Results)

-Lines 82-93: I consider that in this I consider that in this paragraph results/conclusions are combined with objectives which is why I suggest that it would be clearer if the objectives were clearly stated.

Lines 97-98 Four to six naturally healthy or root rot-affected P. notoginseng plants were selected for rhizosphere soil sampling, respectively.

Insert comma.

Lines 100-102: I think that the authors could mention a reference the technique used for the process and conservation of the samples.

Line 104: How was the suspension prepared?

Lines 103-115: 2.2. Isolation and Identification of Fungi. The identification of Trichoderma species is a difficult task.  According to my knowledge, it is not possible to identify the fungal species Trichoderma hamatum only with the primers ITS 1 and ITS 4.

Lines 117-123: I suggest clarified how many isolates/strains of T. hamatum were used in dual culture against the three pathogens.

The authors mention “The dual culture method was used to evaluate antagonistic activity of isolates against the plant pathogens F. oxysporum, F. solani, and F. acutatum on potato dextrose agar (PDA)” and then in the same paragraph: “The antagonistic assay measured pathogen radius and the inhibition zone between the T. hamatum strain and the pathogen.

How many strains or isolates were used in the assay?

How many repetitions were made of each dual combination antagonistic fungus/pathogen?

Lines 128-141: 2.4. Scanning Electron Microscopy

I consider that the objective of electron microscope observation should be clarified.

The pathogenic fungi and T. hamatum were co-cultured on PDA plates as the experimental group. The fungal were cultivated in dual culture?

A 0.5 cm diameter colony…… where it was taken from? From the contact zone?

Lines 142-155: 2.5. DNA Extraction and Sequencing

I consider that the authors could include a sentence clarifying the objective of this study in this part.

Results

Lines 182-185: 3.1. The Antagonistic Activity

The sentence mention “The T. hamatum strain isolated from the rhizosphere soil of healthy P. notoginseng plants demonstrated significant inhibitory effects against F. solani, F. oxysporum, and F. acutatum in dual confrontation experiments”

My comment is in the same sense that I mentioned before. Was only one strain evaluated? If so, based on what criteria was it selected from those previously isolated?

Also, in mat and methods authors mention Lines 97-98: “Four to six naturally healthy or root rot-affected P. notoginseng plants were selected for rhizosphere soil sampling respectively.

I consider that authors should clarified in Mat and Met where the strain used was isolated from (healthy or infected soil).

Lines 317-318: Scanning electron microscopy (SEM) studies revealed that when F. solani was grown alone, its hyphae developed normally without sporulation (Figure. 6A). However, in co- culture conditions, spores were observed adhering to the hyphae (Figure. 6D).

Conidia observed in the figure according to the authors which fungus does it correspond to? See abstract.

Line 328. Fig. 6. SEM images of cultures following inoculation with T. hamatum (D–F). I suggest change this sentence clarifying that are from interaction sections of cocultured fungi

Conclusions: Some comments have already been made before.

Original images/Supplementary materials

The authors include many images of fungal species but they are not clear what they want to show.

Author Response

Manuscript ID: jof-3344698

Title:Unveiling the genomic features and biocontrol potential of Trichoderma hamatum against root rot pathogens

Major comments:

I consider that the work is very interesting because it addresses a very current subject as the search alternatives ecofriendly to reduce the use of agrochemicals to plant disease management. It is important to highlight that the research addresses the study of a Trichoderma species as biological control agents. In this sense, authors evaluated the antagonistic effect of T. hamatum isolated from the rhizosphere soil of Panax notoginseng against threes Fusarium species causing root rot. On the other hand, the authors carried out genomics studies to elucidate the biocontrol mechanisms involved by the T. hamatum which is a contribution of knowledge of this topic. However, I consider that the authors could improve the manuscript by improving and clarifying some aspects of materials and methods. Also, I think the work has some weaknesses in relation to antagonism tests and to the interpretation of the results and the mechanisms involved in the antagonism and they should improve it. For example, it is not clear how many Trichoderma strains were initially isolated from rhizosphere soil. It is also not clear how many strains were used for testing and identification. With respect to the identification, it is widely known that the identification of Trichoderma species is very difficult and that more of ITS regions are required to determine at level species. According to mentioned I recommend its publication in the Journal of Fungi after improvements to the manuscript have been made.

Response:

Thank you for being able to provide valuable comments on the manuscript ID: jof-3344698 “Unveiling the genomic features and biocontrol potential of Trichoderma hamatum against root rot pathogens”, those comments are all valuable and very helpful for revising and improving our paper. We have carefully read the comments and revised the issues. In addition, we have resubmitted a new manuscript, and the modified part is highlighted in red, which is highlighted in blue in the response. If there are any questions in the manuscript, please don't hesitate to let us know. The following sections are our point-by-point responses:

  1. Comment: With respect this sentence: “Scanning electron microscopy revealed that T. hamatum inhibits pathogen growth through sporulating parasitism and secretion of secondary metabolites”. How can the authors can assert the inhibition of the pathogen through secretion of secondary metabolites by scanning electron microscopy? It is not clear the inhibition of the pathogen growth through sporulating parasitism……

Response:

We appreciate your insightful observations. The scanning electron microscopy (SEM) primarily provides morphological evidence, which may not directly confirm biochemical processes such as secondary metabolite secretion. We infer the involvement of secondary metabolites based on:

In the confrontation experiment. we observed clear antibacterial zones, a common indication of secondary metabolite production, which has been previously documented [1,2]. These zones suggest that T. hamatum might be inhibiting pathogen growth through the secretion of secondary metabolites. Additionally, our genomic analysis confirmed the presence of genes associated with secondary metabolite production in T. hamatum. The phenomenon of deformation and dissolution of the mycelial structure was also observed in the SEM images. It is speculated that this was affected by secondary metabolites. Certainly, further biochemical analyses are needed to definitively confirm the involvement of secondary metabolites, which is an area we aim to explore in future studies.

As for “the inhibition of the pathogen growth through sporulating parasitism”, the SEM images presented in Figure 6D and the dual culture experiment Figure 1D illustrate dense sporulation of T. hamatum near the pathogen's hyphae, which correlates with the observed inhibition zones in dual culture assays. While direct quantification or mechanistic studies were not conducted, the phenomenon aligns with the established roles of Trichoderma sporulation in competitive exclusion and biocontrol [3].

We also took into account your suggestion regarding previous misstatements in our manuscript and have revised the relevant sections for clarity. Lines 23-26: “Scanning electron microscopy suggested the hyperparasitism of T. hamatum against F. solani, characterized by spore production that adhered to the pathogen, thereby inhibiting its growth. Furthermore, T. hamatum secreted secondary metabolites that disrupted the mycelial structure and morphology of F. oxysporum and F. acutatum, underscoring its potential as a biocontrol agent.”

Reference:

  1. Atanasova, L.; Knox, B.P.; Kubicek, C.P.; Druzhinina, I.S.; Baker, S.E. The Polyketide Synthase Gene Pks4 of Trichoderma Reesei Provides Pigmentation and Stress Resistance. Eukaryot Cell 2013, 12, 1499–1508, doi:10.1128/EC.00103-13.
  2. Hao, D.; Lang, B.; Wang, Y.; Wang, X.; Liu, T.; Chen, J. Designing Synthetic Consortia of Trichoderma Strains That Improve Antagonistic Activities against Pathogens and Cucumber Seedling Growth. Microb. Cell Factories 2022, 21, 234, doi:10.1186/s12934-022-01959-2.
  3. Mukherjee, P.K.; Mendoza-Mendoza, A.; Zeilinger, S.; Horwitz, B.A. Mycoparasitism as a Mechanism of Trichoderma-Mediated Suppression of Plant Diseases. Fungal Biol. Rev. 2022, 39, 15–33, doi:10.1016/j.fbr.2021.11.004.

  1. Comment: Lines 82-93 I consider that in this paragraph results/conclusions are combined with objectives which is why I suggest that it would be clearer if the objectives were clearly stated.

Response:

Thanks for your suggestion. We have revised the paragraph to separate the objectives, methods, and anticipated contributions of the study. The revised paragraph now explicitly states the study's objective at the beginning, followed by a concise description of the methods used, and concludes with a summary of the expected contributions. Lines 84-96: “This study aimed to investigate the biocontrol potential of T. hamatum against Fusarium species causing root rot in Panax notoginseng. T. hamatum was isolated from the rhizosphere soil of healthy P. notoginseng plants. Specifically, were targeted as the primary pathogens. Plate confrontation assays were conducted to evaluate the differential antagonistic effects of T. hamatum on the targeted Fusarium species. Whole-genome sequencing and comparative genomic analysis were subjected to identify candidate homologous and unique genes. Additionally, scanning electron microscopy and genomic analysis of genes encoding lytic enzymes and secondary metabolites were employed to explore the molecular mechanisms underlying these antagonistic interactions. The findings of this study aim to support the optimization of T. hamatum biocontrol efficacy in managing root rot and contribute to comparative genomic studies of Trichoderma, providing insights into bio-control mechanisms and the evolutionary dynamics of Trichoderma isolates”.

  1. Comment: Lines 97-98 Four to six naturally healthy or root rot-affected P. notoginseng plants were selected for rhizosphere soil sampling, respectively. Insert comma.

Response:

Thanks for your suggestion. We have revised the sentence to enhance clarity and readability. (Lines 100-104)

  1. Comment: Lines 100-102: I think that the authors could mention a reference the technique used for the process and conservation of the samples.

Response:

Thank you for your valuable suggestion. We have provided a referrence to the method of Berendsen, R.L et.al. (2018). We have revised the manuscript to include a reference to the methodology. (Lines 108)

Reference:

  1. Berendsen, R.L.; Vismans, G.; Yu, K.; Song, Y.; De Jonge, R.; Burgman, W.P.; Burmølle, M.; Herschend, J.; Bak-ker, P.A.H.M.; Pieterse, C.M.J. Disease-Induced Assemblage of a Plant-Beneficial Bacterial Consortium. The ISME Journal 2018, 12, 1496–1507, doi:10.1038/s41396-018-0093-1.

  1. Comment: Line 104: How was the suspension prepared?

Response:

Thank you for pointing this out. We adopted the dilution spread plate method for the isolation of fungi. We took 10 g of rhizosphere soil and added it to 90 mL of sterile water for serial dilution.

  1. Comment: Lines 103-115: 2.2. Isolation and Identification of Fungi. The identification of Trichoderma species is a difficult task. According to my knowledge, it is not possible to identify the fungal species Trichoderma hamatum only with the primers ITS 1 and ITS 4.

Response: Thank you for your insightful suggestion. While ITS1/ITS4 primers were initially used for genus-level identification, the species-level identification of T. hamatum in this study was confirmed through whole-genome sequencing and comparative genomic analysis. Phylogenetic analysis based on single-copy orthologous genes and comparative genomic clustering with other Trichoderma species confirmed the strain’s placement within the T. hamatum clade. We also provide a reference of the identified method of Nilsson, R.H. et.al. (2019).

Reference:

  1. Nilsson, R.H.; Anslan, S.; Bahram, M.; Wurzbacher, C.; Baldrian, P.; Tedersoo, L. Mycobiome Diversity: High-Throughput Sequencing and Identification of Fungi. Nat Rev Microbiol 2019, 17, 95–109, doi:10.1038/s41579-018-0116-y.
  2. Comment: Lines 117-123: I suggest clarified how many isolates/strains of T. hamatum were used in dual culture against the three pathogens. The authors mention “The dual culture method was used to evaluate antagonistic activity of isolates against the plant pathogens F. oxysporum, F. solani, and F. acutatum on potato dextrose agar (PDA)” and then in the same paragraph: “The antagonistic assay measured pathogen radius and the inhibition zone between the T. hamatum strain and the pathogen.

Response:

Thank you for your comments. We agree that this requires clarification to ensure the methodology is transparent. In our study, a single strain of T. hamatum was used for the dual culture experiments. This strain was selected based on robust growth and consistent inhibitory effects observed in preliminary screening. We have revised the text accordingly to explicitly state this and standardized the terminology throughout the paragraph. Lines 128-131: “The control was single culture, and the antagonistic experiment was dual culture. The dual culture method was used to evaluate the antagonistic activity of a single T. hamatum strain against the plant pathogens F. oxysporum, F. solani, and F. acutatum on potato dextrose agar (PDA).”

  1. Comment: How many strains or isolates were used in the assay? How many repetitions were made of each dual combination antagonistic fungus/pathogen?

Response:

We appreciate the reviewer’s insightful questions. In this study, only a single strain of T. hamatum was used for the dual culture assays. However, we isolated 120 fungal strains in the preliminary experiments, of which 20 belonged to the genus Trichoderma. The single strain analyzed in the study was selected based on its strong antagonistic activity observed during preliminary screening. For each dual combination of T. hamatum and the pathogens (F. oxysporum, F. solani, and F. acutatum), three independent biological replicates were conducted. Additionally, each biological replicate included three technical replicates, ensuring statistical robustness. We have revised the “Materials and Methods” section to include these details for clarity. Lines 131-134: “Three independent biological replicates were conducted for each pathogen-antagonist combination, with each replicate performed in triplicate. The antagonistic assay measured the pathogen radius and the inhibition zone between the T. hamatum strain and the pathogen.”

  1. Comment: Lines 128-141: 2.4. Scanning Electron Microscopy: I consider that the objective of electron microscope observation should be clarified. The pathogenic fungi and T. hamatum were co-cultured on PDA plates as the experimental group. The fungal were cultivated in dual culture? A 0.5 cm diameter colony…… where it was taken from? From the contact zone?

Response:

Thank you for your valuable comments. We clarifyed the objectives of SEM observations We have revised the manuscript to address these concerns. Lines 141-143: “The objective of using SEM was to examine the interactions between T. hamatum and the pathogens, focusing on structural changes in pathogen hyphae and evidence of hyperparasitism or inhibition.”

The experimental group consisted of dual culture plates with T. hamatum and the pathogens (F. oxysporum, F. solani, and F. acutatum) co-cultured on PDA. The photo of T. hamatum cultured alone has been added to the Supplementary Material as Figure S7 for your review.

The 0.5 cm diameter colony samples were specifically collected from the interaction zone where T. hamatum and pathogen mycelia overlapped. This region was chosen to investigate the direct interactions between the antagonist and the pathogens. Lines 143-146: “The three pathogenic fungi were cultured on potato dextrose agar (PDA) plates, serving as the control group (CK). For the experimental group, T. hamatum and the pathogenic fungi were co-cultured on PDA plates. A 0.5 cm diameter colony sample, including agar, was excised from the interaction zone where T. hamatum and pathogen mycelia overlapped.”

  1. Comment: Lines 142-155: 2.5. DNA Extraction and Sequencing

I consider that the authors could include a sentence clarifying the objective of this study in this part.

Response:

Thank you for your suggestion. We have revised the text to include a statement linking this methodology to the study’s goals in Lines 159-160: “The purpose of DNA extraction and sequencing was to analyze the genomic features of T. hamatum that contribute to its antagonistic activity and biocontrol potential against root rot pathogens.”

  1. Comment: Lines 182-185: 3.1. The Antagonistic Activity

The sentence mention “The T. hamatum strain isolated from the rhizosphere soil of healthy P. notoginseng plants demonstrated significant inhibitory effects against F. solani, F. oxysporum, and F. acutatum in dual confrontation experiments”

My comment is in the same sense that I mentioned before. Was only one strain evaluated? If so, based on what criteria was it selected from those previously isolated?

Also, in mat and methods authors mention Lines 97-98: “Four to six naturally healthy or root rot-affected P. notoginseng plants were selected for rhizosphere soil sampling respectively.

I consider that authors should clarified in Mat and Met where the strain used was isolated from (healthy or infected soil).

Response:

Thank you for your thoughtful comments. We appreciate the opportunity to provide clarification. We isolated 120 fungal strains in the preliminary experiments, of which 20 belonged to the genus Trichoderma. This strain was selected based on its strong antagonistic activities against the three pathogenic fungi observed during the preliminary screening. T. hamatum was isolated from healthy soil. I have already made the revisions in the manuscript. Lines 100-104: “The plot chosen for sampling was land where P. notoginseng had been continuously cultivated, and no fungicides were used during the growth period, allowing the plants to grow naturally. Among the large area of P. notoginseng plants affected by root rot, we picked out those that remained healthy as our samples. Each replicate consisted of 4 to 6 plants, and three replicates were conducted.”

  1. Comment: Lines 317-318: Scanning electron microscopy (SEM) studies revealed that when F. solani was grown alone, its hyphae developed normally without sporulation (Figure. 6A). However, in co- culture conditions, spores were observed adhering to the hyphae (Figure. 6D).

Conidia observed in the figure according to the authors which fungus does it correspond to? See abstract.

Response:

Thank you for highlighting this observation. The spores observed adhering to the hyphae in co-culture conditions (Figure 6D) are produced by T. hamatum. As noted in our study, F. solani does not sporulate under the given experimental conditions when cultured alone (Figure 6A). Moreover, our genomic analysis identified the presence of the pks4 gene in T. hamatum as reported by Zeilinger, S et.al. (2016). In T. reesei, the pks4 gene has been shown to be an ortholog of the pigment-forming polyketide synthase (PKS) responsible for the biosynthesis of ulosonic acid and bicoumarin in Fusarium species. Functional studies have demonstrated that deletion of the pks4 gene in T. reesei directly impacts green conidial pigmentation, exosporium structure, conidial cell wall stability, and antagonistic activity against other fungi. This suggests that the observed conidia originate from T. hamatum during co-culture. I have already made the revisions in the manuscript. Lines 340-343: “Scanning electron microscopy (SEM) studies revealed that when F. solani was grown alone, its hyphae developed normally without sporulation (Figure 6A). However, in co-culture conditions, conidia of T. hamatum were observed adhering to F. solani hyphae in the interaction zone (Figure 6D).”

Reference:

  1. Atanasova, L.; Knox, B.P.; Kubicek, C.P.; Druzhinina, I.S.; Baker, S.E. The Polyketide Synthase Gene Pks4 of Trichoderma Reesei Provides Pigmentation and Stress Resistance. Eukaryot Cell 2013, 12, 1499–1508, doi:10.1128/EC.00103-13.

  1. Comment: Line 328. Fig. 6. SEM images of cultures following inoculation with T. hamatum (D–F). I suggest change this sentence clarifying that are from interaction sections of cocultured fungi.

Response:

Thank you for your valuable suggestion. We have revised the figure legend to explicitly state that these images correspond to the interaction zone of co-cultured T. hamatum and pathogenic fungi. Liens 352-355: “SEM images of cultures following inoculation with T. hamatum (D–F). Show images from the in-teraction zone of co-cultured T. hamatum and pathogenic fungi, highlighting conidia adherence to pathogen hyphae and structural changes associated with fungal antagonism.”

  1. Comment: Conclusions: Some comments have already been made before.

Response:

Thank you for your observation regarding the Conclusions section. We have carefully reviewed the comments provided earlier in the review process and have revised the Conclusions. Liens 450-462: “This study underscores the significant antagonistic properties of T. hamatum against key Fusarium pathogens causing root rot in P. notoginseng, demonstrating its potential applications in sustainable agriculture. Through comprehensive whole-genome sequencing and analysis, we identified critical genomic features, in-cluding genes encoding carbohydrate-active enzymes and secondary metabolite bio-synthetic clusters, which are central to T. hamatum’s biocontrol efficacy. Scanning elec-tron microscopy further revealed two primary inhibitory mechanisms: sporulating parasitism and the secretion of secondary metabolites, which effectively disrupted pathogen hyphal structures. These findings enhance our understanding of the molec-ular and mechanistic basis of T. hamatum’s biocontrol activity and provide a theoretical framework for developing innovative biological control strategies for P. notoginseng cultivation. By integrating genomic insights and experimental evidence, this research lays the groundwork for environmentally friendly disease management solutions, fos-tering sustainable agricultural practices.”

  1. Comment: Original images/Supplementary materials

The authors include many images of fungal species but they are not clear what they want to show.

Response:

Thank you for your comment regarding the scanning electron microscopy (SEM) images. As per the journal's requirements, we have submitted the original, high-resolution SEM images to ensure transparency and accuracy. These images provide clear visualization of the interactions between Trichoderma hamatum and the Fusarium pathogens, as described in the manuscript.

Reviewer 3 Report

Investigación sobre el uso de Trichoderma hamatum para el control de Fusarium sp. interés actual, ya que el uso del control biológico con estos especies de hongos, es una estrategia prometedora, sostenible y natural que contribuye a reducir, complementar o sustituir el uso de agroquímicos. Se presentan hallazgos importantes que contribuyen a mejorar la comprensión de los mecanismos de control biológico de T. hamatum, además, se evidencian para determinar las posibles aplicaciones de T. hamatum en Producción de alimentos saludables.

El La investigación del genoma de T. hamatum ayuda a comprender las bases del antagonismo fúngico mecanismos contra las especies de Fusarium que causan la pudrición de la raíz en P. notoginseng y es de gran importancia, teniendo en cuenta que la base científica de la mecanismos antagonistas de Trichoderma contra especies de Fusarium Completamente aclarado.

Author Response

Manuscript ID: jof-3344698

Title: Unveiling the genomic features and biocontrol potential of Trichoderma hamatum against root rot pathogens

Major comments:

Investigación sobre el uso de Trichoderma hamatum para el control de Fusarium sp. interés actual, ya que el uso del control biológico con estos especies de hongos, es una estrategia prometedora, sostenible y natural que contribuye a reducir, complementar o sustituir el uso de agroquímicos. Se presentan hallazgos importantes que contribuyen a mejorar la comprensión de los mecanismos de control biológico de T. hamatum, además, se evidencian para determinar las posibles aplicaciones de T. hamatum en Producción de alimentos saludables.

El La investigación del genoma de T. hamatum ayuda a comprender las bases del antagonismo fúngico mecanismos contra las especies de Fusarium que causan la pudrición de la raíz en P. notoginseng y es de gran importancia, teniendo en cuenta que la base científica de la mecanismos antagonistas de Trichoderma contra especies de Fusarium Completamente aclarado.

Response:  

Thank you for being able to provide valuable comments on the manuscript ID: jof-3344698 “Unveiling the genomic features and biocontrol potential of Trichoderma hamatum against root rot pathogens”, those comments are all valuable and very helpful for revising and improving our paper. We have carefully read the comments and revised the issues. In addition, we have resubmitted a new manuscript, and the modified part is highlighted in red, which is highlighted in blue in the response. If there are any questions in the manuscript, please don't hesitate to let us know. The following sections are our point-by-point responses:

  1. Comments :Lines 97-98: What was the phenological stage of the plants? How was the representative sample size of the plot with P. notoginseng plants determined?

Response:

We selected samples of three-year-old Panax notoginseng that were in the fruiting stage. The plot chosen for sampling was land where P. notoginseng had been continuously cultivated, and no fungicides were used during the growth period, allowing the plants to grow naturally. Among the large area of P. notoginseng plants affected by root rot, we picked out those that remained healthy as our samples. Each replicate consisted of 4 to 6 plants, and three replicates were conducted. Therefore, we have revisions to lines 99-104 of the “Soil samples were collected from the Wenshan Miaoxiang P. notoginseng Technology Experimental Field in October 2022. The plot chosen for sampling was land where P. notoginseng had been continuously cultivated, and no fungicides were used during the growth period, allowing the plants to grow naturally. Among the large area of P. notoginseng plants affected by root rot, we picked out those that remained healthy as our samples. Each replicate consisted of 4 to 6 plants, and three replicates were conducted. The plants were carefully excavated to preserve root system integrity, and loose soil was gently shaken off. Adhering rhizosphere soil was then gently brushed off. Samples were immediately placed in iceboxes for preservation and stored at -80°C upon arrival at the laboratory[31].”

Reference:

  1. Berendsen, R.L.; Vismans, G.; Yu, K.; Song, Y.; De Jonge, R.; Burgman, W.P.; Burmølle, M.; Herschend, J.; Bak-ker, P.A.H.M.; Pieterse, C.M.J. Disease-Induced Assemblage of a Plant-Beneficial Bacterial Consortium. The ISME Journal 2018, 12, 1496–1507, doi:10.1038/s41396-018-0093-1.

  1. Comments: Lines 105-106: What technique of isolation and purification of the fungus was used?

Response:

Thank you for pointing this out. We adopted the dilution spread plate method for the isolation of fungi. Distinct fungal colonies were picked and purified by sub-culturing onto fresh PDA plates under sterile conditions. Pure cultures were obtained by hyphal tip trnsfer and inoculation until contamination-free fungal growth was confirmed. Therefore, we have revisions to lines 107-111 of the “We employed the dilution spread plate method for fungal isolation. To prepare a serial dilution, 10 grams of rhizosphere soil were mixed with 90 milliliters of sterile water. The resulting soil suspension was diluted to 10⁻⁵, and 100 μL of the diluted solution was inoculated onto potato dextrose agar (PDA) plates. Distinct fungal colonies were subsequently picked and purified by sub-culturing onto fresh PDA plates under sterile conditions. Pure cultures were obtained through hyphal tip transfer and repeated inoculations until contamination-free fungal growth was confirmed. ”

  1. Comments : Lines 119-120: How many replicates were used in the antagonist assay? what design experimenatl was used?

Response:

Thank you for your insightful comments. In our experiment, each antagonist assay was conducted with three biological replicates to ensure statistical robustness. The number of replicates was determined based on established protocols for antagonist assays in fungal biocontrol studies [1], ensuring sufficient statistical power.I have already made the revisions in the manuscript. Liens 129-131: “The control was single culture, and the antagonistic experiment was dual culture. The dual culture method was used to evaluate the antagonistic activity of a single T. hamatum strain against the plant pathogens F. oxysporum, F. solani, and F. acutatum on potato dextrose agar (PDA). Three independent biological replicates were conducted for each pathogen-antagonist combination, with each replicate performed in triplicate. The antagonistic assay measured the pathogen radius and the inhibition zone between the T. hamatum strain and the pathogen.”

Reference:

  1. Zhou, X.; Wang, J.; Liu, F.; Liang, J.; Zhao, P.; Tsui, C.K.M.; Cai, L. Cross-Kingdom Synthetic Microbiota Supports Tomato Suppression of Fusarium Wilt Disease. Nat. Commun. 2022, 13, 7890, doi:10.1038/s41467-022-35452-6.

  1. Comments: Lines 122-123: Were plates with the antagonist fungus used as control?

Response:

Thank you for your valuable question. In the confrontation experiment, we used individual antagonistic fungi as controls. I have included this as Figure S7 in the supplementary materials.

  1. Comments: Lines 124-126: was the category or class of antagonistic fungi against pathogenic fungi measured? Was the ability of sporulation of the antagonist on the pathogenic fungi measured?

Response:

Thank you for your insightful comments. In the dual culture assays, the grading was determined by the reduction in the radius of the pathogen and the size of the inhibition area. Nevertheless, during the preliminary screening of antagonistic strains, T. hamatum displayed a more potent antagonistic capacity in contrast to other strains. And we have already stated in Result 1 that "The inhibition rates of T. hamatum against these three Fusarium species were 68.07%, 70.63%, and 66.12%, respectively". Our experiment solely measured the sporulation ability of T. hamatum under solitary culture conditions. The spore concentration was 5.44 × 10⁸ spores/mL when cultured alone for 7 days.

  1. Comments: What statistical analysis was carried out on the data obtained?

Response:

Thank you for your observation. The details of the statistical analysis conducted for fungal experiments have been provided in the “Materials and Methods”” section under the subsection “2.8 Statistical Analysis”. Lines 199-202: “For the antagonistic activity assays. All experiments were conducted in triplicate to ensure reproducibility, and the results are presented as mean ± standard deviation (SD). Statistical analyses were performed using SPSS v27.0.”

  1. Comments: Table S2? Table S3? Table S4? Table S5?

Response: Thanks for your careful checks. I have made the revisions in the manuscript, specifically at lines 206, 209, 213, and 229.

  1. Comments: Lines 343-345: … and environmental remediation (reference to literature could be included)

Response:Thank you for your suggestion to include supporting references for the statement regarding T. hamatum’s role in environmental remediation. We agree that providing literature support strengthens the claim. We have revised the text to include relevant references.(Liens 371)

  1. Comments:Lines 356-357: In addition, the abundant sporulation of Trichoderma allows rapid dissemination and colonization in the ecological niche, where it is competing with other microorganisms

Response: Thank you for your thoughtful comments. We fully agree with your view that spores contribute to the colonization and competitive ability of biocontrol fungi. and the revised manuscript now includes further details on this topic in the relevant sections (please refer to lines 383-385).

Reviewer 4 Report

In this work, the genome of a strain of Trichoderma hamatum isolated from the roots of a plant was sequenced and analyzed. This strain inhibited the mycelial growth of three Fusarium species. Genome sequencing of this fungus allowed the identification of genes codifying to CAZyme involved in fungal cell wall degradation. In addition, biosynthetic gene clusters (BGCs) codifying enzymes that participate in the synthesis of secondary metabolites were identified.

 General concept comments

The manuscript has significant deficiencies and conceptual errors. The conclusions are not based on the results. Many sentences are just speculations.

  • Specific comments 

Introduction:

Lane 55 to 57: Secondary metabolite produced by Trichoderma are not well described.

Materials and Methods

Statistical analyses are not included

Results

Antagonistic activity. In this section, authors did not mention the number of replicate.

Lane 189, what does mean Parasitic sporulation?

Lane 192-193. Results do not show secretion of secondary metabolites

Lane 304-306. This phrase has important conceptual errors

Lane 314. Figure 5 is confusing

Lane 321. Is figure 6 B and C?

Discussion, there are many conceptual errors (For example peramine and clavariac acid, synthesized in the biosynthetic gene clusters of T. hamatum……), speculative phrases and conclusions not based on results

Author Response

Manuscript ID: jof-3344698

Title: Unveiling the genomic features and biocontrol potential of Trichoderma hamatum against root rot pathogens

Major comments:

In this work, the genome of a strain of Trichoderma hamatum isolated from the roots of a plant was sequenced and analyzed. This strain inhibited the mycelial growth of three Fusarium species. Genome sequencing of this fungus allowed the identification of genes codifying to CAZyme involved in fungal cell wall degradation. In addition, biosynthetic gene clusters (BGCs) codifying enzymes that participate in the synthesis of secondary metabolites were identified.

Response:

Thank you for being able to provide valuable comments on the manuscript ID: jof-3344698 “Unveiling the genomic features and biocontrol potential of Trichoderma hamatum against root rot pathogens”, those comments are all valuable and very helpful for revising and improving our paper. We have carefully read the comments and revised the issues. In addition, we have resubmitted a new manuscript, and the modified part is highlighted in red, which is highlighted in blue in the response. If there are any questions in the manuscript, please don't hesitate to let us know. The following sections are our point-by-point responses:

  1. General concept comments: The manuscript has significant deficiencies and conceptual errors. The conclusions are not based on the results. Many sentences are just speculations.

Response:

Thank you for your detailed feedback. We apologize for all unclear expressions. We have carefully reviewed the manuscript and made substantial revisions to address the deficiencies, clarify conceptual errors, and align the conclusions more scrupulously with the results.

  1. Comment: Line 55 to 57: Secondary metabolite produced by Trichoderma are not well described.

Response:

We appreciate your valuable feedback. In response, we have thoroughly revised this section to enhance the description of Trichoderma secondary metabolites based on published literature. We have revised the sentence to enhance clarity and readability in Lines 57-60 in the main text: “Trichoderma-derived secondary metabolites include non-ribosomal peptides, such as peptaibiotics, siderophores, and diketopiperazines (e.g., gliotoxin and gliovirin), as well as polyketides, terpenes, pyrones, and isocyanide metabolites [18].”

Reference:

  1. Zeilinger, S.; Gruber, S.; Bansal, R.; Mukherjee, P.K. Secondary Metabolism in Trichoderma – Chemistry Meets Genomics. Fungal Biology Reviews 2016, 30, 74–90, doi:10.1016/j.fbr.2016.05.001.

  1. Comment: Materials and Methods: Statistical analyses are not included

Response:

Thank you for your observation. The details of the statistical analysis conducted for fungal experiments have been provided in the “Materials and Methods”” section under the subsection “2.8 Statistical Analysis”. Lines 199-202: “For the antagonistic activity assays. All experiments were conducted in triplicate to ensure reproducibility, and the results are presented as mean ± standard deviation (SD). Statistical analyses were performed using SPSS v27.0.”

  1. Comment: Antagonistic activity. In this section, authors did not mention the number of replicates.

Response:

Thank you for your insightful comments. In our experiment, each antagonist assay was conducted with three biological replicates to ensure statistical robustness. We have already made the revisions in the manuscript. Lines 131-132: “Three independent biological replicates were conducted for each pathogen-antagonist combination, with each replicate performed in triplicate.”

  1. Comment: Line 189, what does mean Parasitic sporulation?

Response:

Thank you for highlighting this point. We recognize that the term “parasitic sporulation” requires clarification to ensure its meaning is explicit and accessible to the readers. Therefore, we have revisions to lines 209-211: “Notably, in the confrontation with F. solani, T. hamatum mycelium completely overgrew the F. solani strain, leading to the production of numerous green spores, indicating that T. hamatum parasitizes F. solani and produces spores during this process. (Figure. 1D).”

  1. Comment: Line 192-193. Results do not show secretion of secondary metabolites

Response:

We appreciate your comments. The secretion of secondary metabolites by T. hamatum was speculated by the inhibition zone observed in the dual culture method. We agree that the evidence for secondary metabolite secretion requires further clarification. Therefore, we have revisions to lines 215-217 of the “This suggests that T. hamatum secretes secondary metabolites that are toxic to F oxysporum and F. acutatum, inhibiting their growth, with a more pronounced effect on F. oxysporum.”

  1. Comment: Lane 304-306. This phrase has important conceptual errors.

Response:

Thank you for bringing this to our attention. We recognize that the phrasing in this sentence may have led to conceptual ambiguity. Therefore, we have revised the text accordingly to explicitly state this and standardized the terminology throughout the paragraph. Lines 317-334: “A total of 52 biosynthetic gene clusters (BCGs) distributed across eight contigs were predicted (Figure S6). Among these, 26 BCGs related to non-ribosomal peptide synthetases (NRPS and NRPS-like), 16 BCGs encoded gens related to type I polyketide synthases (T1PKS), and 7 BCGs were terpene synthases. To assess the biocontrol potential of T. hamatum, BGCs associated with eleven types of secondary metabolites were identified through genome mining (Table S8). The structural formulas of secondary metabolites and their genetic region are shown in Figure 5. Three genes exhibited 100% amino acid sequence homology with known secondary metabolic secretion gene clusters were further analyzed. The region 2.6 of Ptg2 encodes NRPS genes involved in the synthesis of peramine (Figure 5j). The region 6.5 of Ptg6 encodes terpene-related genes related to the synthesis of clavaric acid (Figure 5k). Region 5. 2 in Ptg5 were NRPS-like gene and involved in the synthesis of choline (Figure 5g). Additionally, two genes display 75% amino acid sequence similarity. The genes in region 1.6 (Pgt1, NRPS) related to trichoxide synthesis (Figure 5a), and region 4.3 (Pgt4, T1PKS,) related to synthesis of dimerumic acid 11-mannoside, and dimerumic acid (Figure 5e, f). Other biosynthetic gene clusters (BGCs) are responsible for the biosynthesis of aurofusarin (Figure 5b), melinacidin IV (Figure 5c), leucinostatin A (Figure 5d), leucinostatin B (Figure 5h), and BII-rafflesfungin (Figure 5i).

  1. Comment: Lane 314. Figure 5 is confusing

Response:

Thank you for your valuable feedback regarding Figure 5. We understand that its current presentation may cause confusion. Therefore, we have revised Figure 5 and the corresponding paragraph in Lines 323-336.

  1. Comment: Lane 321. Is figure 6 B and C?

Response:

Thanks for your careful checks. We are sorry for our careless. We have revised “Figure 6A, B” in the original text to “Figure 6B, C”. (Lines 345)

  1. Comment: Discussion, there are many conceptual errors (For example peramine and clavariac acid, synthesized in the biosynthetic gene clusters of T. hamatum……), speculative phrases and conclusions not based on results.

Response:

Thank you for your detailed feedback. We appreciate your efforts in identifying areas for improvement in the discussion section. After careful review, we have made significant changes to address the issues of conceptual inaccuracies, speculative phrasing, and unsupported conclusions. Here are the specific revisions. Lines 410-413: “Furthermore, genome mining suggests that biosynthetic gene clusters in T. hamatum may be involved in the synthesis of peramine and clavariac acid, compounds that are commonly associated with insecticidal activity [51,52].”

Reference:

  1. Krasnoff, S.B.; Keresztes, I.; Gillilan, R.E.; Szebenyi, D.M.E.; Donzelli, B.G.G.; Churchill, A.C.L.; Gibson, D.M. Serinocyclins A and B, Cyclic Heptapeptides from Metarhizium Anisopliae. J. Nat. Prod. 2007, 70, 1919–1924, doi:10.1021/np070407i.
  2. Nelli, M.R.; Scheerer, J.R. Synthesis of Peramine, an Anti-Insect Defensive Alkaloid Produced by Endophytic Fungi of Cool Season Grasses. J. Nat. Prod. 2016, 79, 1189–1192, doi:10.1021/acs.jnatprod.5b01089.

Round 2

Reviewer 1 Report

The Trichoderma hamatum genome FBL 587 is publicy available at the genetic sequence database GenBank (NCBI):

https://www.ncbi.nlm.nih.gov/nuccore/SEIV00000000.1

LOCUS       SEIV01000000            1803 rc    DNA     linear   PLN 24-AUG-2021

DEFINITION  Trichoderma hamatum strain FBL 587, whole genome shotgun sequencing

            project.

ACCESSION   SEIV00000000

VERSION     SEIV00000000.1

DBLINK      BioProject: PRJNA513966

            BioSample: SAMN10717514

KEYWORDS    WGS.

SOURCE      Trichoderma hamatum

  ORGANISM  Trichoderma hamatum

            Eukaryota; Fungi; Dikarya; Ascomycota; Pezizomycotina;

            Sordariomycetes; Hypocreomycetidae; Hypocreales; Hypocreaceae;

            Trichoderma.

REFERENCE   1  (bases 1 to 1803)

  AUTHORS   Davolos,D., Russo,F., Canfora,L., Malusa,E., Tartanus,M.,

            Furmanczyk,E.M., Ceci,A., Maggi,O. and Persiani,A.M.

  TITLE     A Genomic and Transcriptomic Study on the DDT-Resistant Trichoderma

            hamatum FBL 587: First Genetic Data into Mycoremediation Strategies

            for DDT-Polluted Sites

  JOURNAL   Microorganisms 9 (8), 1680 (2021)

  REMARK    DOI: 10.3390/microorganisms9081680

REFERENCE   2  (bases 1 to 1803)

  AUTHORS   Davolos,D.

  TITLE     Direct Submission

  JOURNAL   Submitted (31-JAN-2019) Department of Technological Innovations and

            Safety of Plants, Products and Anthropic Settlements, INAIL-DIT,

            Research Area, Via R. Ferruzzi 38/40, Rome 00143, Italy

COMMENT     The Trichoderma hamatum whole genome shotgun (WGS) project has the

            project accession SEIV00000000.  This version of the project (01)

            has the accession number SEIV01000000, and consists of sequences

            SEIV01000001-SEIV01001803.

            ##Genome-Assembly-Data-START##

            Assembly Date          :: 2018

            Assembly Method        :: SPAdes v. 3.11

            Genome Representation  :: Full

            Expected Final Version :: No

            Genome Coverage        :: 44.0x

            Sequencing Technology  :: Illumina MiSeq

            ##Genome-Assembly-Data-END##

FEATURES             Location/Qualifiers

     source          1..1803

                     /organism="Trichoderma hamatum"

                     /mol_type="genomic DNA"

                     /strain="FBL 587"

                     /isolation_source="polluted soil"

                     /db_xref="taxon:49224"

                     /geo_loc_name="Poland"

                     /collection_date="2017"

WGS         SEIV01000001-SEIV01001803

//

https://www.ncbi.nlm.nih.gov/Traces/wgs/SEIV01?display=download

GenBank:SEIV01.1.gbff.gz16.6 Mb

FASTA:SEIV01.1.fsa_nt.gz11.6 Mb

ASN.1:SEIV01.1.bbs.gz9.4 Mb

The Trichoderma hamatum genome FBL 587 is publicy available at the genetic sequence database GenBank (NCBI):

https://www.ncbi.nlm.nih.gov/nuccore/SEIV00000000.1

LOCUS       SEIV01000000            1803 rc    DNA     linear   PLN 24-AUG-2021

DEFINITION  Trichoderma hamatum strain FBL 587, whole genome shotgun sequencing

            project.

ACCESSION   SEIV00000000

VERSION     SEIV00000000.1

DBLINK      BioProject: PRJNA513966

            BioSample: SAMN10717514

KEYWORDS    WGS.

SOURCE      Trichoderma hamatum

  ORGANISM  Trichoderma hamatum

            Eukaryota; Fungi; Dikarya; Ascomycota; Pezizomycotina;

            Sordariomycetes; Hypocreomycetidae; Hypocreales; Hypocreaceae;

            Trichoderma.

REFERENCE   1  (bases 1 to 1803)

  AUTHORS   Davolos,D., Russo,F., Canfora,L., Malusa,E., Tartanus,M.,

            Furmanczyk,E.M., Ceci,A., Maggi,O. and Persiani,A.M.

  TITLE     A Genomic and Transcriptomic Study on the DDT-Resistant Trichoderma

            hamatum FBL 587: First Genetic Data into Mycoremediation Strategies

            for DDT-Polluted Sites

  JOURNAL   Microorganisms 9 (8), 1680 (2021)

  REMARK    DOI: 10.3390/microorganisms9081680

REFERENCE   2  (bases 1 to 1803)

  AUTHORS   Davolos,D.

  TITLE     Direct Submission

  JOURNAL   Submitted (31-JAN-2019) Department of Technological Innovations and

            Safety of Plants, Products and Anthropic Settlements, INAIL-DIT,

            Research Area, Via R. Ferruzzi 38/40, Rome 00143, Italy

COMMENT     The Trichoderma hamatum whole genome shotgun (WGS) project has the

            project accession SEIV00000000.  This version of the project (01)

            has the accession number SEIV01000000, and consists of sequences

            SEIV01000001-SEIV01001803.

            ##Genome-Assembly-Data-START##

            Assembly Date          :: 2018

            Assembly Method        :: SPAdes v. 3.11

            Genome Representation  :: Full

            Expected Final Version :: No

            Genome Coverage        :: 44.0x

            Sequencing Technology  :: Illumina MiSeq

            ##Genome-Assembly-Data-END##

FEATURES             Location/Qualifiers

     source          1..1803

                     /organism="Trichoderma hamatum"

                     /mol_type="genomic DNA"

                     /strain="FBL 587"

                     /isolation_source="polluted soil"

                     /db_xref="taxon:49224"

                     /geo_loc_name="Poland"

                     /collection_date="2017"

WGS         SEIV01000001-SEIV01001803

//

https://www.ncbi.nlm.nih.gov/Traces/wgs/SEIV01?display=download

GenBank:SEIV01.1.gbff.gz16.6 Mb

FASTA:SEIV01.1.fsa_nt.gz11.6 Mb

ASN.1:SEIV01.1.bbs.gz9.4 Mb

Author Response

Manuscript ID: jof-3344698

Title: Unveiling the genomic features and biocontrol potential of Trichoderma hamatum against root rot pathogens

Dear Reviewer,1:

Thank you for being able to provide valuable comments on the manuscript ID: jof-3344698 “Unveiling the genomic features and biocontrol potential of Trichoderma hamatum against root rot pathogens”, those comments are all valuable and very helpful for revising and improving our paper. We have carefully read the comments and revised the issues. In addition, we have resubmitted a new manuscript, and the modified part is highlighted in red, which is highlighted in blue in the response. If there are any questions in the manuscript, please don't hesitate to let us know.

Major comments:

The Trichoderma hamatum genome FBL 587 is publicy available at the genetic sequence database GenBank (NCBI): https://www.ncbi.nlm.nih.gov/nuccore/SEIV00000000.1

GenBank: SEIV01.1.gbff.gz16.6 Mb

FASTA: SEIV01.1.fsa_nt.gz11.6 Mb

ASN.1: SEIV01.1.bbs.gz9.4 Mb

Response:

Thank you very much for providing the data and information. We have successfully located the genome data for Trichoderma hamatum FBL 587 in the National Center for Biotechnology Information (NCBI). We have re-conducted the comparative genomic analysis, and based on the results of this analysis, we have made the necessary revisions to the original text. Lines 262-295: “The phylogenetic relationship of the T. hamatum strain was evaluated based on Internal Transcribed Spacer (ITS) sequences, comparing it with ten other Trichoderma species and two outgroup fungi. The analysis utilized 2,378 shared single-copy ho-mologous sequences, providing a robust framework for assessing genetic similarities and evolutionary relationships (Figure. 3). The ITS sequence clusters T. atroviride, T. gamsii, T. hamatum XP3S-3oo, T. hamatumX2zz, T. asperellum, and T. hamatum into a sin-gle branch. Similarly, single-copy homologous sequences group T. gamsii, T. atroviride, T. asperellum, T. hamatum GD12, T. hamatum FBL587 and T. hamatum into one branch. The phylogenetic tree that the T. hamatum strain forms a distinct clade closely related to T. hamatum GD12 and T. hamatum FBL587 strains, confirming its taxonomic place-ment within the T. hamatum species. Analysis of orthologous relationships among strains revealed that no paralogous genes were detected in the T. hamatum strain, and the number of genes not involved in clustering was minimal. Furthermore, its gene structure closely resembles that of the T. hamatum GD12 strain reported in previous studies. (Figure S3). These findings suggest that the T. hamatum strain has undergone relatively stable evolution, with highly specialized gene functions.

Phylogenetic trees were constructed using the eleven Trichoderma species and two outgroup strains. The average divergence time for the outgroup represented by Aspergillus fumigatus is estimated to be 187.8 million years ago (MYA). This predates the di-versification of Hypocreaceae and Xylariaceae. Within the Trichoderma clade, the di-vergence from Xylaria occurred approximately 141.7 MYA. Within Hypocreaceae, the split between T. hamatum and other species occurred around 7.1 MYA. T. hamatum diverged from T. hamatum FBL587 and T. hamatum GD12 relatively recently, around 0.9 million years ago, strains T. hamatum FBL587 and T. hamatum GD12, di-verged relatively recently 0.3 MYA, suggested rapid evolution within this subgroup. Specifically, the strain of T. hamatum examined in this study diverged from T. hamatum GD12 approximately 0.9 MYA (Figure. S4). During the evolution of the involved thir-teen fungal species, gene family contractions occurred more frequently than gene fam-ily expansions. The Trichoderma species, including T. virens and T. harzianum, demon-strated a higher level of gene family expansion 186 and 174, respectively compared to other species in the clade. Notably, T. hamatum experienced significant evolutionary changes in 332 gene families, comprising 163 expansions and 196 contractions (Figure. S5).”

Reviewer 4 Report

The manuscript still contains important conceptual errors.

The conclusions are not based on the results.

Many sentences are just speculations.

Authors must correct conceptual errors throughout the manuscript, but mainly in the discussion.

Author Response

Manuscript ID: jof-3344698

Title: Unveiling the genomic features and biocontrol potential of Trichoderma hamatum against root rot pathogens

Dear Reviewer,4:

Thank you for being able to provide valuable comments on the manuscript ID: jof-3344698 “Unveiling the genomic features and biocontrol potential of Trichoderma hamatum against root rot pathogens”, those comments are all valuable and very helpful for revising and improving our paper. We have carefully read the comments and revised the issues. In addition, we have resubmitted a new manuscript, and the modified part is highlighted in red, which is highlighted in blue in the response. If there are any questions in the manuscript, please don't hesitate to let us know. The following sections are our point-by-point responses:

Major comments:

The manuscript still contains important conceptual errors.

The conclusions are not based on the results.

Many sentences are just speculations.

Response:

Thank you for your insightful comments. We have carefully examined the points raised and revised the conclusion. The changes are highlighted in blue font in the paper and can be found on line 455-466. “This study isolated a T. hamatum strain with antagonistic properties against Fusarium oxysporum, Fusarium solani, and Fusarium acutatum, the pathogens responsible for root rot in P. notoginseng. Whole-genome sequencing revealed that the T. hamatum strain possesses 10,774 genes and is most closely related to the previously reported T. hamatum GD12 and T. hamatum FBL587. Antagonistic assays and scanning electron mi-croscopy suggested that T. hamatum employs sporulating parasitism against F. solani and hyphal lysis against the other two pathogens. Additionally, genes associated with conidiospore pigmentation, carbohydrate-active enzymes, and secondary metabolite biosynthetic clusters were identified, supporting the strain’s antagonistic properties. These findings enhanced our understanding of the molecular and mechanistic basis of T. hamatum's biocontrol activity and provide a theoretical framework for developing biological control strategies in P. notoginseng cultivation.”

Detail comments:

Authors must correct conceptual errors throughout the manuscript, but mainly in the discussion.

Response: Thank you for your thoughtful comments. Regarding the conceptual errors in the discussion, we have initiated a comprehensive review. We have made significant revisions to the discussion part.

Lines 358-383: “Biological control, as an environmentally friendly approach to disease prevention and management, reduces reliance on chemical pesticides and mitigates the environmental pollution associated with their use. It has been widely applied in crop cultivation. However, root rot disease caused by Fusarium in the continuous cropping of P. notoginseng, a traditional Chinese medicinal plant, remains a significant challenge. This study investigated the antagonistic strain T. hamatum, isolated from the rhizosphere soil of P. notoginseng, for its potential to combat root rot disease. The strain demonstrated significant antagonistic activity against the primary pathogens responsible for P. notoginseng root rot, including F. oxysporum, F. solani, and F. acutatum (Table S1). Trichoderma plays a significant role in global agricultural production and serves as an important resource for biological control[15,34].

This study analyzed the genomic structure of biocontrol fungus T. hamatum through genome sequencing and conducted a comparative genomic analysis with ten other Trichoderma species, as well as two outgroup species. The phylogenetic tree revealed that T. hamatum is most closely related to T. hamatum GD12 and T. hamatum FBL587, confirming its taxonomic placement within the T. hamatum species (Figure 3). The observed gene family expansions (163 genes) and contractions (196 genes) suggest significant genomic restructuring, potentially contributing to ecological adaptation and functional specialization. The absence of paralogous genes and minimal unclustered genes further indicates genomic stability. Paralogous genes are a subclass of homologous genes generated through gene duplication[38]. Their formation represents a critical mechanism for enhancing genetic diversity during genome evolution. The previously studied T. hamatumGD12 strain has been well-documented for its effectiveness in promoting plant growth (PGP) and its strong biological control capabilities[39]. The T. hamatum strain analyzed in this study exhibits a high degree of similarity to T. hamatum GD12 in gene structure (Figure S3), suggesting its significant potential for biological control applications.”

Lines 387-404: “The direct biological control mechanisms employed by Trichoderma against plant pathogens include parasitism, antibiotic production, the action of lytic enzymes, competition for ecological niches, and resource competition[15,35]. These molecular mechanisms are associated with multiple genes in Trichoderma[36]. The plate confrontation assay and scanning electron microscopy demonstrated that T. hamatum employed distinct strategies to inhibit pathogenic fungi causing root rot in various Fusarium species. The antagonistic mode against F. solani and F. acutatum involved parasitism, with F. solani being targeted through sporogenic parasitism and F. acutatum through puncture parasitism. In contrast, F. oxysporum was inhibited via hyphal lysis, highlighting a different mode of antagonism. The inhibition zones observed on the plate suggest that T. hamatum likely produces antibiotics or specific metabolites that inhibit the growth of the pathogenic fungi F. oxysporum and F. acutatum(Figure 1 and 6). Moreover, the abundant spores of Trichoderma enable it to disperse efficiently and establish dominance within ecological niches, thereby enhancing its competitive ability against other microorganisms[37].The green spores are potentially linked to the polyketide synthase (PKS) encoding genes in Trichoderma. Pigment-forming PKSs play a dual role in conidiospore pigmen-tation and the biosynthesis of low molecular weight pigments, such as aurofusarin and bikaverin[44].Low molecular weight pigment-forming PKSs are implicated in fungal defense, me-chanical stability, and stress resistance[45,46].”

Lines 410-418: “Therefore, PKS-encoding genes potentially play a crucial role in the antagonistic activity against pathogens. Genomic functional annotation, antagonistic experiments, and SEM analysis all suggest that T. hamatum has the potential to suppresses pathogen growth through parasitism.

Fungal secondary metabolite biosynthetic gene clusters (BGCs) are responsible for synthesizing a wide array of secondary metabolites with specific biological functions[37]. In the genome of T. hamatum, a total of 11 BGCs were identified, which are associated with the production of various antibiotic and antibacterial compounds, including trichoxide, melinacidin IV, leucinostatin A/B, peramine, and clavariac acid.”

Lines 423-426: “The significant inhibitory effect of T. hamatum on F. oxysporum and F. acutatum may be attributed to the secondary metabolites it produces. Furthermore, BGCs in T. hamatum also involved in the synthesis of peramine and clavariac acid, compounds that are commonly associated with insecticidal activity[45,46].”

Lines 432-436: “The action of lytic enzymes in the CAZymes analysis of the T. hamatum genome, 28 genes were detected in glycoside hydrolase family 18 (GH18) and 8 genes in family 75 (GH75) (Table S7). GH18 encompasses all fungal chitinases, which are enzymes that hydrolyze glycosidic bonds in chitin. Chitosanases from GH75 are involved in chitin degradation [48].”

Lines 446-453: “Besides, The analysis of T. hamatum also identified other enzymes, including β-glucosidase, β-galactosidase, α-galactosidase, and exo-β-1,3-glucanase (Table S7). The chitinases, β-glucosidases, and mannosidases secreted by Trichoderma work together to control pathogens like Rhizoctonia solani, and Fusarium spp.[57,58]. The hyperparasitic response of T. harzianum ALL42 to pathogens is host-dependent, with variations in hyphal entanglement and secreted proteins observed[57]. Overall, the biocontrol mechanisms of Trichoderma can vary significantly depending on the specific host it interacts with.”

Round 3

Reviewer 4 Report

The manuscript still contains important conceptual errors.

The conclusions are not based on the results.

Many sentences are just speculations.

The manuscript still contains important conceptual errors.

The conclusions are not based on the results.

Many sentences are just speculations.

Author Response

Manuscript ID: jof-3344698

Title: Unveiling the genomic features and biocontrol potential of Trichoderma hamatum against root rot pathogens

Dear Reviewer,4:

Thank you for being able to provide valuable comments on the manuscript ID: jof-3344698 “Unveiling the genomic features and biocontrol potential of Trichoderma hamatum against root rot pathogens”, those comments are all valuable and very helpful for revising and improving our paper. We carefully reviewed the comments, thoroughly checked the manuscript, and revised the introduction, materials and methods, results, and conclusion sections of the article. In addition, we have resubmitted a new manuscript, and the modified part is highlighted in red, which is highlighted in blue in the response. If there are any questions in the manuscript, please don't hesitate to let us know. The following sections are our point-by-point responses:

Major comments:

The manuscript still contains important conceptual errors.

The conclusions are not based on the results.

Many sentences are just speculations.

Response:

Thank you for your insightful comments. We have carefully examined the points raised and revised the conclusion. The changes are highlighted in blue font in the paper and can be found on

Introduction

lines 36-37. “Chemical residues can cause environmental pollution, pose risks to human health, and contribute to pathogen resistance.”

Lines 41-42: “However, the molecular and mechanistic basis of fungal suppression in biological con-trol strategies remains poorly understood.

Lines 44-46: “Among the promising biocontrol agents, species of the genus Trichoderma have gained attention for their ability to suppress soil-borne pathogens and promote plant health.”

Lines 48-49: “Trichoderma, with its ability to utilize a wide range of nutrient sources, can successfully establish itself as a parasitic or multifunctional symbiotic organism.”

Lines 54-58: “Parasitism is a typical behavior of Trichoderma, which produces hydrolytic enzymes to degrade the cell walls of host fungi, thereby parasitizing ascomycete fungi or closely related fungal species[19]. The production of secondary metabolites in Trichoderma include antibiotic production and other non-ribosomal peptides, such as peptaibiotics, diketopiperazines, polyketides, and so on[18].”

Lines 61-63: “Trichoderma species, such as T. harzianum ZC51 and Trichoderma AAUW1, effectively control plant diseases caused by Fusarium, with the latter achieving an 86.67% inhibi-tion rate against F. solani

Lines 70-77: “Despite belonging to the same genus, the substantial genetic, phenotypic, and ecological diversity among Trichoderma species is reflected in their genomic variability[26]. Whole-genome sequencing of species such as T. harzianum, T. asperellum, and T. atroviride has enhanced our understanding of Trichoderma's ecological and genetic characteristics, enabling comprehensive molecular analyses of biocontrol mechanisms[27–30]. Understanding the genomic structure of Trichoderma aids in elucidating the genetic and molecular mechanisms of fungal antagonism against Fusarium species that cause root rot in P. notoginseng.”

Material and Methods

lines 161-163. “After library preparation, sequencing was performed on the PacBio Sequel II system using the SMART sequencing method.”

Lines 164-168: “The longest high-quality genomic regions were identified using SMRT Link (version 7.0) software and the HQRF (High-Quality Region Finder) tool. Low-quality regions were filtered based on Signal-to-Noise Ratio (SNR). Reads exceeding 1000 bp, a thresh-old ensuring sufficient read length for downstream analyses, were selected as quali-ty-controlled sequencing data.

Lines 173-174: “The initial genome assembly was performed using Hifiasm (version 0.13) on third-generation sequencing data.”

Lines 173-174: “To evaluate assembly quality, genome completeness was assessed with BUSCO (ver-sion 5.2.2) based on the fungi_odb10 database.”

Lines 180-182: “RepeatMasker (version open-1.0.11) was used to identify transposable element (TE) repeat sequences in T. hamatum using a curated repeat sequence database. ”

Lines 189-191: “Gene functions were annotated by aligning genome protein sequences to the Non-Redundant Protein Database (NR), KEGG, Swiss-Prot, and KOG databases using Blastp (version 2.9.0).”

Results

Lines 201-208: “In its confrontation with F. solani, T. hamatum completely overgrew the pathogen's mycelium and produced numerous green spores, which may suggest a parasitic interaction, although the direct mechanism remains to be confirmed (Figure. 1D). For F. oxysporum and F. acutatum, inhibition zones were observed, appearing transparent and diffuse, respectively (Figure. 1E, F). These findings indicate that T. hamatum likely employs multiple mechanisms, including the potential secretion of antifungal compounds, to inhibit these pathogens.”

Lines 215-222: “HiFi sequencing of the T. hamatum was performed, generating 6.82 Gb of raw data with an average sequencing depth of 162.6×. The sequencing produced 393,002 high-quality reads after filtering (Table S2). The Hi-C-assisted genome assembly resulted in a total genome length of 41.93 Mb, an N50 value of 6.07 Mb, and 31 contigs. The GC content of the assembled genome was 46.37% (Figure. 2, Table S3). Genome alignment showed an alignment rate of 99.00%, with a coverage of 99.98%, indicating a high-quality assembly. BUSCO analysis further demonstrated that 98.7% of core conserved genes were complete, highlighting the genome’s completeness and integrity for downstream analyses (Table S4).”

Lines 231-243: “Annotation of non-coding RNAs (ncRNAs) revealed 400 distinct ncRNAs, including 221 tRNAs, 338 rRNAs, and 25 snRNAs (Table 1).

Gene annotation was conducted using InterPro, GO, KEGG, SwissProt, TrEMBL, and NR database (Table S6). Functional annotation through the GO database classified 7,922 genes (73.53% of all genes) into three main categories: biological processes (6,557 genes), cellular components (5,130 genes), and molecular functions (9,303 genes). Among these categories, genes involved in cellular and metabolic processes were the most prevalent. Molecular functions constituted the largest category,  with primary roles including catalytic activity, protein binding, and transporter activity (Figure. S1). KEGG pathway analysis identified 4,164 genes (38.65%) mapped to five major pathways, with metabolism-related genes being the most abundant, particularly those involved in carbon, amino acid, and lipid metabolism (Table S6). Additionally, 37 genes were identified as being involved in the biosynthesis of secondary metabolites (Figure. S2).”

Lines 289-295: “Additionally, 8 genes from the GH55 family encode exo-β-1,3-glucanase, and 6 genes from the GH75 family encode chitosanase, further emphasizing the hydrolases synesis function of the fungi (Table S7). These enzymes facilitate the breakdown of structural components in fungal cell walls, supporting the parasitic activity of T. hamatum. Collectively, the presence of these hydrolase genes highlights the significant potential of T. hamatum as a mycoparasitic biocontrol agent.”

Lines 324-334: “Scanning electron microscopy (SEM) revealed distinct morphological responses of pathogenic fungi in co-culture with T. hamatum. When F. solani was grown alone, its hyphae appeared smooth and intact, with no evidence of sporulation (Figure 6A). In co-culture conditions, T. hamatum conidia were observed adhering to F. solani hyphae in the interaction zone (Figure 6D). Similarly, in the absence of the antagonistic strain, the hyphae of F. oxysporum and F. acutatum were smooth and regular (Figure 6B, C). However, in the presence of T. hamatum, F. oxysporum hyphae exhibited pronounced dissolution and lysis (Figure 6E), while F. acutatum hyphae developed irregular wrinkles and interwoven mycelial masses (Figure 6F). These findings indicate that T. hamatum employs distinct inhibitory mechanisms against the three pathogenic fungi, as evidenced by the varied morphological alterations observed.”

Conclusion

Lines 439-450: “This study isolated a T. hamatum strain with significant antagonistic activity against F. oxysporum, F. solani, and F. acutatum, the pathogens responsible for root rot in P. notoginseng. Whole-genome sequencing revealed a complete genome with 10,774 genes, including carbohydrate-active enzymes (e.g., GH18 chitinases) and secondary metabolite biosynthetic clusters (e.g., aurofusarin and melinacidin IV), highlighting its biocontrol potential. Antagonistic assays demonstrated inhibition rates of 70.63%, 68.07%, and 66.12% against F. solani, F. oxysporum, and F. acutatum, respectively. Scanning electron microscopy revealed distinct inhibitory mechanisms, including sporulating parasitism against F. solani and hyphal lysis against F. oxysporum and F. acutatum. These findings provide specific insights into the genetic and mechanistic basis of T. hamatum's biocontrol activity, offering a robust foundation for developing sustainable biological control strategies in P. notoginseng cultivation.”
